# MARKED INDUCING POINT CASCADED SDEs FOR NEURAL MANIFOLD LEARNING

## ABSTRACT

The manifold hypothesis suggests that high-dimensional neural time series lie on a low-dimensional manifold shaped by simpler underlying dynamics. To uncover this structure, latent dynamical variable models such as state-space models, recurrent neural networks, neural ordinary differential equations, and Gaussian process latent variable models are widely used. We propose MIP-CSDE (Marked Inducing Point Cascaded Stochastic Differential Equations), a novel cascaded stochastic differential equation model that balances computational efficiency with interpretability and addresses key limitations of existing approaches. Our model assumes that a sparse set of trajectory samples suffices to reconstruct the underlying smooth manifold. The manifold dynamics are modeled using a set of Brownian bridge SDEs, with points – specified in both time and value–drawn from a multivariate marked point process. These Brownian bridges define the drift of a second set of SDEs, where their trajectories are mapped to the observed data. This cascade model structure enables continuous, differentiable latent processes that can model arbitrarily complex time series as the number of inducing points increases. For MIP-CSDE, we derive training and inference procedures, and demonstrate that computational complexity of its inference step scales as $\mathcal{O}(P \cdot N)$, exhibiting linear dependence on the observation data length $N$, where $P$ is the number of particles. We then show its application in both synthetic data and neural recordings, where our proposed model accurately recovers the underlying manifold structure and scales effectively with data dimensionality.

## 1 INTRODUCTION

The manifold hypothesis proposes that high-dimensional neural time series lie on a low-dimensional manifold shaped by simpler latent dynamics (Whiteley et al., 2024). Evidence for such structure can be found in auditory cortex activity (Bondanelli et al., 2021) and in speech signals constrained by vocal tract mechanics (Gonzalez-Castillo et al., 2023). Methods for uncovering latent manifolds include state-space models (SSMs) (Särkkä and Svensson, 2023), dynamical autoencoders (Girin et al., 2020), switching SSMs (Ghahramani and Hinton, 2000), Gaussian and Dirichlet processes (Fox et al., 2008; Eleftheriadis et al., 2017; Wang et al., 2005), t-SNE and UMAP (Van der Maaten and Hinton, 2008; McInnes et al., 2018), and Latent Neural ODEs (Rubanova et al., 2019). In this paper, we focus on SSMs for high-dimensional neural time series. Classical models include Linear Gaussian SSMs (Kitagawa and Gersch, 1996) and Hidden Markov Models (Rabiner, 2002), while modern variants include Deep SSMs (Rangapuram et al., 2018), Deep Kalman Filters (Krishnan et al., 2015a), Gaussian Process Dynamical Models (GPDMs) (Wang et al., 2005), Gaussian Process SSMs (GPSSMs) (Eleftheriadis et al., 2017), and nonlinear latent-variable models such as Latent Factor Analysis via Dynamical Systems (LFADS) (Sussillo et al., 2016) and Gaussian Process Factor Analysis (GPFA) (Yu et al., 2008). A critical challenge is to develop robust and computationally efficient inference and training procedures, where recent approaches such as SDE Inference via Natural Gradients (SING) (Hu et al., 2025)improve inference for latent SDEs. Despite these advances, limitations remain: some models fail to capture oscillatory dynamics, others require structural constraints, and deep neural network (DNN) based approaches are data-hungry. Sequential models like recurrent neural networks (RNNs) and long short-term memory networks (LSTMs) (Chang et al., 2024) capture nonlinear dependencies but pose interpretability and training challenges (Glorot and Bengio, 2010). To address these limitations, we propose MIP-CSDE which balances interpretability and expressive power. Inspired by findings that neural manifolds evolve smoothly along low-dimensional trajectories (Cunningham and Yu, 2014a; Gosztolai et al., 2023), our proposed model assumes that a sparse set of trajectory samples suffices to reconstruct the mani-

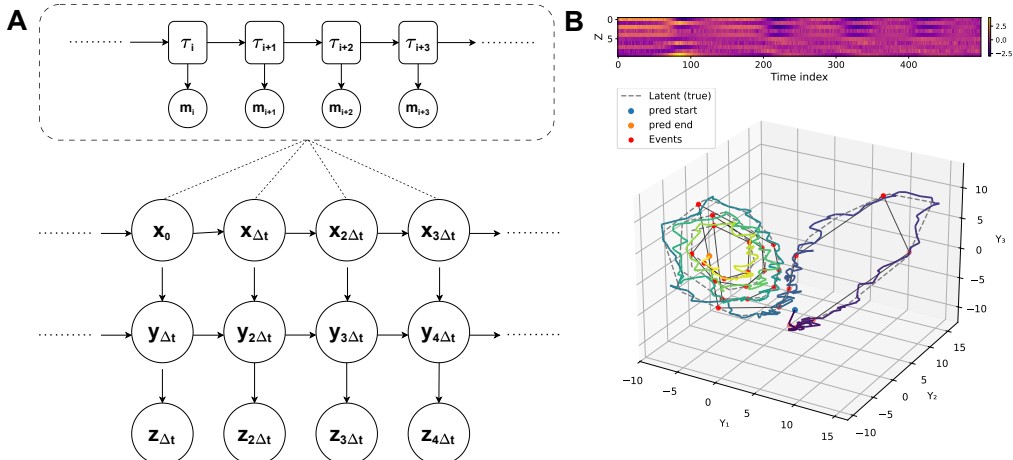

Figure 1: **MIP-CDSE Model Structure and its Application in Inferring Lorenz System Trajectory:** (A) Shows the graphical representation of our proposed model, where inducing points $(\tau_i, m_i)$ go through two layers of SDEs $(X_t, Y_t)$ followed by projection to the observation domain $(Z_k)$. (B) Top row shows the observed data used as input to our proposed model. This data is generated by projecting the Lorenz trajectory into a 10-dimensional observation space, where the mapping is defined by a linear projection with additive multivariate Gaussian noise. The bottom row shows the underlying Lorenz trajectory and its estimation by our model, along with the timing of the inducing points. MIP-CSDE simultaneously learns the mapping from the manifold to the observed space and infers the trajectory of the Lorenz attractor. The timing and values of the inducing points reflect their adaptive behavior in capturing both fast and slow transitions along the trajectory.

fold. The first layer generates trajectories with Brownian bridge SDEs, using inducing points from a multivariate marked point process (Daley and Vere-Jones, 2006; Oksendal, 2013). These trajectories define the drift for a second set of SDE layer, whose outputs map to observed data. This cascaded structure yields a continuous, differentiable latent process capable of modeling arbitrarily complex dynamics as inducing points increase. We derive efficient inference and training procedures, with their computaitonal complexity scale linearly with data length. We then show that MIP-CSDE robustly and accurately recovers the latent manifolds in both synthetic and neural datasets. Figure 1 illustrates the MIP-CSDE graphical model and its application to synthetic data generated using the Lorenz dynamics.

The paper is organized as follows. Section 2 presents MIP-CSDE, its properties, and inference procedure. Section 3 shows model application in simulated and neural data for manifold inference and decoding tasks. Section 4 and 5 respectively cover the discussion and conclusion sections.

## 2 MATERIALS & METHODS

Neural population activity is widely observed to evolve along low-dimensional latent manifolds whose trajectories exhibit smooth temporal structure Cunningham and Yu (2014b). Motivated by this observation, we propose a cascade SDE framework combined with inducing points spanning both the time and value spaces to characterize neural population dynamics and their relationship to an underlying smooth manifold. In this section, we first define the components of our model, including the generation of inducing points and the SDEs that map these points to the observed time series data. We then establish the universal approximation properties of the proposed model and develop its training and inference procedures.

### 2.1 CASCADE SDE AND INDUCING POINT MOTIVATION

Our objective is to construct a generative model capable of producing trajectories that are continuous and smooth, i.e., with well-defined temporal derivatives. Standard diffusion processes or single-layer SDEs typically produce sample paths that are continuous but nowhere differentiable, making them ill-suited for modeling smooth neural trajectories.

To generate flexible continuous paths, we first employ a Brownian-bridgebased SDE in which inducing points define a piecewise-linear interpolation, enabling the construction of complex continuous functions. However, such trajectories are not smooth. To obtain differentiable paths, we introduce a second SDE whose drift is driven by the trajectory of the first SDE. By pushing the noise variance of this second SDE toward zero, the resulting trajectories converge to smooth, differentiable functions while preserving the flexibility provided by the inducing-point structure.

In this cascade system, the first SDE with inducing points offers a piecewise-linear approximation of any continuous function, while the second SDE ensures that the overall trajectory is smooth and differentiable, aligning with the smooth manifold structure underlying neural population activity.

## 2.2 CASCADE SDE FRAMEWORK

Figure 1A illustrates the model structure. The inducing points are a set of event-value pairs that sample the underlying manifold in both time and value space. Each event is characterized by a time $t_i$ and an associated mark vector $\vec{m}_i$. The model considers an arbitrary finite sequence of these pairs, represented as the set $\mathcal{I} = \{(\vec{m}_i, t_i); \; i = 1, 2, \dots\}$. The joint probability distribution of event-value pairs over a period $T$ for $L$ events is given by:

$$p\Big(\{(t_i, \vec{m}_i)\}_{i=1}^{L}, T\Big) = \exp\left(-\int_0^T \lambda(t \mid \mathcal{H}_t)\, dt\right) \prod_{i=1}^{L} \lambda(t_i \mid \mathcal{H}_{t_i})\, p(\vec{m}_i \mid t_i, \mathcal{H}_{t_i}) \tag{1}$$

where $\lambda(t_i \mid \mathcal{H}_{t_i})$ is the event occurrence rate conditioned on the history of previous events $\mathcal{H}_i$, and $p(\vec{m}_i \mid t_i, \mathcal{H}_{t_i})$ defines the mark distribution conditioned on the event time $t_i$ and the history $\mathcal{H}_i$ (Jacobsen, 2006). The sequence of events can also be described using waiting times, defined as $\tau_i = t_i - t_{i-1}$, which transforms the process into a renewal marked point process (Daley and Vere-Jones, 2006).

With the inducing points generated, we now introduce the remaining components of the model that map these points to the observed data. Let $Z_k \in \mathbb{R}^M$ denote the observed data at discrete time points $k = 1, \dots, K$, which are modeled as functions of an underlying continuous latent process $Y_t \in \mathbb{R}^D$, where $D \ll M$. $Y_t$ evolves according to another latent process $X_t$, which has the same dimension as $Y_t$. The latent process $X_t = \{x_t^d\}_{d=1}^D$ is modeled using a set of SDEs, where the process is constrained to reach the mark values at times specified by the inducing time points - i.e, a Brownian bridge SDE (Pitman and Yor, 1999). Evolution of state process in each dimension $d = 1, \dots, D$ is defined by:

$$dx_t^d = \mu_t^d\, dt + \sigma_t^d\, dw_t^d, \tag{2}$$

where $w_t^d$ is a standard Wiener process, $\mu_t^d$ is the drift term, and $\sigma_t^d$ is the time-dependent diffusion coefficient. For $t \in [t_i, t_{i+1})$, which corresponds to the waiting period $\tau_{i+1}$, the drift and diffusion terms are defined as:

$$\mu_t^d = \frac{m_{i+1}^d - x_t^d}{t_{i+1} - t}, \quad \sigma_t^d = \sqrt{\frac{(t_{i+1} - t)(t - t_i)}{t_{i+1} - t_i}}, \tag{3}$$

where $m_{i+1}^d$ is the $d$-th component of the mark vector $\vec{m}_{i+1}$, the value that the process must reach at the event time $t_{i+1}$. The latent process $Y_t = \{y_t^d\}_{d=1}^D$ evolves according to:

$$dy_t^d = (-\zeta\, y_t^d + x_t^d)\, dt + \sigma_y^d\, d\nu_t^d, \zeta > 0 \tag{4}$$

where the drift term for $y_t^d$ is defined by $x_t^d$, $\nu_t^d$ is a standard Wiener process, and $\sigma_y^d$ is the diffusion coefficient for the $d$-th component. where we generate $\sigma_y^d = 0$, which guarantees first-order differentiability (no sharp jump or edge). We generally set $\zeta$ toward 1, but it can be set to smaller values or trained. Finally, the observations $Z_k$ are defined as:

$$Z_k = W \begin{pmatrix} y_{k\Delta t}^1 \\ \vdots \\ y_{k\Delta t}^D \end{pmatrix} + \varepsilon_k, \tag{5}$$

where $\Delta t$ denotes the sampling interval at which the observations $Z_k$ are collected, $W \in \mathbb{R}^{M \times D}$ is a linear projection matrix, and $\varepsilon_k \sim \mathcal{N}(0, R)$ represents Gaussian noise with covariance $R \in \mathbb{R}^{M \times M}$. Here, we assume a linear projection with additive noise; more generally, the framework can accommodate more complex and non-linear mappings. For instance, for the hippocampus data analyzed in Section 3.3, the mapping between $Y_t$ and $Z_k$ is a non-linear function characterizing the rate function for a point-process observation.

## 2.3 MODEL PROPERTIES

In this section, we discuss two key attributes of MIP-CSDE: its universal approximation capability and its computational cost. Other aspects of the model, including its nonparametric nature and strategies for managing the growth of inducing points, are discussed in the Appendix A.1.

### 2.3.1 UNIVERSAL APPROXIMATION PROPERTY

When the process is deterministic and the inducing points are equally spaced, we can rely on the sampling theorem which suggests that a signal can be completely reconstructed from its samples (Shannon, 1949). In simple terms, any continuous function can be reconstructed from properly sampled data points. Here, we extend a similar idea to the cascade SDEs using the inducing points.

**Theorem:** Let $f \in C([0, T])$ be a continuous function and let $\varepsilon > 0$ be arbitrary. Then there exists a choice of inducing points such that the expected one-dimensional component of the process, $\mathbb{E}[s_t]$, uniformly approximates the integral

$$g(t) = \int_0^t f(s)\, ds \tag{6}$$

within error $\varepsilon$, in the sense that

$$\sup_{t \in [0,T]} |\mathbb{E}[s_t] - g(t)| < \varepsilon. \tag{7}$$

Here, $s_t$ denotes the $d$-th component $y_t^d$ of the process $Y_t = \{y_t^d\}_{d=1}^D$.

**Proof:** In our model, each component is given by $s_t = \int_0^t x_s^d\, ds + \sigma W_t^d$, where $x_t^d$ is a Brownian bridge, $W_t^d$ is a standard Brownian motion, and $\sigma$ is the noise variance.

Let $\{t_i\}_{i=1}^N$ be uniformly spaced inducing points, each associated with a mark $m_i$. Define $x_t^d$ by piecewise linear interpolation of these inducing points. Since piecewise linear functions are dense in $C([0, T])$, we can choose $\{m_i\}$ such that $x_t^d \to f(t)$ uniformly on $[0, T]$. Define $s_t = \int_0^t x_s^d\, ds$. Because integration preserves uniform convergence, it follows that

$$\mathbb{E}[s_t] \to g(t) = \int_0^t f(s)\, ds \quad \text{uniformly on } [0, T] \tag{8}$$

The noise term satisfies $\mathbb{E}[\sigma W_t^d] = 0$ and $\mathrm{Var}(\sigma W_t^d) = \sigma^2 t$. Applying Chebyshevs inequality, we have $P(|s_t - \mathbb{E}[s_t]| \geq \eta) \leq \frac{\sigma^2 T}{\eta^2}$, so for sufficiently small $\sigma$, the process $s_t$ concentrates around its expectation. In Appendix A.2, we establish that as $N \to \infty$, the spacing of the inducing points converges to $T/N$. Thus, by selecting appropriate inducing points $\{(t_i, m_i)\}$, we ensure

$$\sup_{t \in [0,T]} |\mathbb{E}[s_t] - g(t)| < \varepsilon \tag{9}$$

**Corollary:** Let $Y_t \in \mathbb{R}^D$ be a continuous vector-valued function on $[0, T]$. For any $\varepsilon > 0$, there exists a choice of inducing points and model parameters such that each component of the MIP-CSDE latent process $Y_t^{(\text{model})}$ satisfies

$$\sup_{t \in [0,T]} \left\| \mathbb{E}\left[ Y_t^{(\text{model})} \right] - Y_t \right\|_2 < \varepsilon \tag{10}$$

In particular, any continuous multidimensional time series that can be expressed as a continuous transformation of such a latent process (e.g., through Eq. (5)) can be approximated arbitrarily well by MIP-CSDE, for suitable latent dimension and inducing-point density.

**Proof:** This corollary extends the one-dimensional approximation result to continuous multidimensional trajectories. It does not specify a unique manifold representation, but shows that such a representation can be constructed given sufficient latent dimensionality and a suitably dense set of inducing points. We can assume that each dimension of $Y_t$ is independent of the others; thus, we only need to adjust the corresponding inducing point values to capture each specific dynamic. Note that as the number of sample points increases, convergence to the bound is achieved using the same set of inducing point times across dimensions.

This proof does not specify the manifold representation but shows that any such representation can be constructed with suitable dimensions and a sparse set of inducing points. In practice, marks and times are adjusted from the observations, creating dependencies across dimensions and among the inducing points.

### 2.3.2 COMPUTATIONAL COST

A key advantage of the proposed model lies in its favorable computational complexity compared to other non-parametric models such as GPs. Standard GPs require inversion of an $N \times N$ covariance matrix in their prediction step, which results in a computational complexity of $\mathcal{O}(N^3)$ (where N is the number of samples or time points) (Seeger, 2004). This scaling substantially restricts the applicability of GP for long temporal sequences or high-frequency data. In contrast, as we show in the next section, inference for the discrete representation of our model can be performed using a sequential Monte Carlo (SMC) (Doucet et al., 2001) approach. When using particle-based methods such as Particle Marginal Metropolis-Hastings (PMMH) Andrieu et al. (2010), the computational cost of trajectory inference scales as $\mathcal{O}(P \cdot N)$, where $P$ is the number of particles and $N$ again denotes the number of time points. This linear scaling with respect to $N$ enables our proposed model to handle long trajectories more efficiently, making it well suited for characterization of high-resolution neural data. Moreover, the computational cost of inference is independent of the number of inducing points, since neither their number nor their values affect the SMC procedure, which will be discussed next. While generating inducing points incurs additional computational cost, their sampling rate is adapted to the underlying dynamics and scales as $O(L)$, where $L$ is the maximum number of inducing points.

### 2.4 MODEL TRAINING AND INFERENCE

The training objective is to maximize the marginal likelihood (or evidence) of the observed data $\{Z_k\}_{k=0}^K$. This requires updating multiple sets of parameters, which include the event–value posterior distributions, and inferring the trajectories of the latent processes $X_t$ and $Y_t$ over $t \in [0, T]$. For this purpose, we use an Expectation–Maximization (EM) algorithm, which naturally accommodates latent processes (Brown and Kass, 2018). A variational inference alternative (Blei et al., 2017) is discussed in Appendix A.3.

To develop the EM algorithm, we first derive a discrete-time representation of the model. This representation is obtained using the renewal waiting-time process introduced in Section 2.1. Under this assumption, the non-Markovian dependence of $X_t$ is removed, which is a critical modeling step for deriving recursive inference methods such as SMC. Intuitively, at time $t$, the next inducing point and its mark are already known, which eliminates dependence of the current $X_t$ and $Y_t$ on the future trajectory of $X_t$.

Within the discrete representation, the times of the inducing points (events) are defined in continuous space; thus, the trajectories of $X_t$ and $Y_t$ can be reconstructed at any temporal resolution. Appendix A.4 provides additional details on discrete SDE approximations. With the discrete representation of MIP-CSDE and the renewal process for the inducing points, the full likelihood of the model is given by

$$P(Z_{1:K}, X_{0:K}, Y_{0:K}, \tau_{1:n_u}, \vec{m}_{1:n_u}; \Omega, \Psi) = p(X_0, Y_0) \prod_{k=1}^{K} \Big[ p(Z_k \mid Y_k, W, R) \, p(Y_k \mid X_{k-1}, Y_{k-1}, \{\sigma_y^d\}_{d=1}^D)$$

$$\times \, p(X_k \mid X_{k-1}, \tau_{1:n_s}, \{\sigma_x^d\}_{d=1}^D, \vec{m}_{1:n_s}) \Big] \times \prod_{n=1}^{n_u} \Big[ p(\tau_n \mid H_n) \, p(\vec{m}_n \mid \tau_n, H_n) \Big] p_e \, p(\Psi) \quad (10.\text{a})$$

$$n_s = \min\{n : \tau_n > k\} \quad (10.\text{b})$$

The parameter set is $\Omega = \{W, R, \{\sigma_y^d\}_{d=1}^D, \{\sigma_x^d\}_{d=1}^D, \{\mu_{0,x}^d, \sigma_{0,x}^d\}_{d=1}^D, \{\mu_{0,y}^d, \sigma_{0,y}^d\}_{d=1}^D\}$, representing the SDE and observation-model parameters. The set $\Psi$ contains the parameters describing the time-value distribution. We assume the waiting times $\tau_i$ are independent of previous events and follow a Gamma distribution. The marks $\vec{m}_i$ are assumed independent of previous events and of the current waiting time, and follow a multivariate distribution with zero mean and diagonal covariance. Thus, $\Psi = \big\{\{\mu_m^d, \sigma_m^d, \alpha^d, \beta^d\}_{d=1}^D\big\}$, where $\mu_m^d, \sigma_m^d$ are the mark-distribution parameters and $\alpha^d, \beta^d$ are the Gamma-distribution parameters.

The quantity $n_u$ denotes the number of events in the interval $[0, T]$. The term $p_e$ is the likelihood contribution for the non-event period after the final event $n_u$ until time $T$, given by

$$p_e = \prod_{d=1}^{D} \exp\left( - \int_0^{T - t_{n_u}} f(s; \alpha^d, \beta^d) \, ds \right) \quad (11)$$

---

**Algorithm 1** SMC Algorithm for Inferring Inducing Points and State Estimation

---

1: **Set Algorithm Hyperparameters:**
2: Set number of particles $U$
3: Define initial distributions $p(x_0)$ and $p(y_0)$
4: Define proposal density $\pi_k(\cdot)$
5: Set hyperparameters $\alpha_0, \beta_0$ for $\tau$ distribution
6: Set hyperparameters $\mu_0, \xi_0$ for $m$ distribution

7: **Initialization:**
8: **for** $u = 1$ to $U$ **do**
9:     Sample $x_0^u \sim p(x_0), y_0^u \sim p(y_0)$        ▷ Draw initial latent variables
10:     Set $m_0^u = \vec{0}, \tau_0^u = 0, n_u = 0$        ▷ Initialize inducing-point memory
11:     Set initial weight $w_k^u = \frac{1}{U}$        ▷ Uniform particle weights
12:     Initialize particle $D_0^u = \{x_0^u, y_0^u, \tau_0^u, m_0^u, n_u\}$        ▷ Store complete particle state
13: **end for**

14: **Inference:**
15: **for** $k = 1$ to $K$ **do**
16:     **1. Time & Mark Sampling:**
17:     **for** $u = 1$ to $U$ **do**
18:        **if** $k \cdot \Delta t > \tau_{\max(n_u)}^u$ **then**
19:           Sample $\tau_{\text{new}}^u \sim \Gamma(\tau; \alpha_0, \beta_0)$        ▷ Sample a new inducing-point time
20:           Sample $m_{\text{new}}^u \sim \mathcal{N}(m; \mu_0, \xi_0)$        ▷ Sample corresponding inducing-point mark
21:           Update $D_k^u = \{\dots\}$        ▷ Append inducing point to particle history
22:        **end if**
23:     **end for**
24:     **2. Sampling:**
25:     **for** $u = 1$ to $U$ **do**
26:        Sample $(x_k^u, y_k^u) \sim \pi_k(\cdot)$        ▷ Draw new latent states from proposal
27:        Compute importance weight:

$$w_k^u = w_{k-1}^u \cdot \frac{p(z_k \mid y_k^u)\, p(y_k^u \mid x_{k-1}^u)\, p(x_k^u \mid \tau_{0:n_u}^u, m_{0:n_u}^u)}{\pi_k(x_k^u, y_k^u \mid x_{0:k-1}^u, y_{0:k-1}^u, z_k, \tau_{0:n_u}^u, m_{0:n_u}^u)}$$

            ▷ Correct for discrepancy between proposal and true posterior

28:     **end for**
29:     **3. Normalization:**
30:     **for** $u = 1$ to $U$ **do**

$$\hat{w}_k^u = \frac{w_k^u}{\sum_{v=1}^{U} w_k^v}$$

            ▷ Normalize weights to obtain posterior over particles

31:     **end for**
32:     **4. Resampling:**
33:     Resample $U$ particles $D_k^u$ with probabilities $\hat{w}_k^u$        ▷ Mitigate particle degeneracy
34:     **for** $u = 1$ to $U$ **do**
35:        Reset weight: $w_k^u = \frac{1}{U}$        ▷ Equalize weights after resampling
36:     **end for**
37: **end for**

---

where $f(\cdot; \alpha^d, \beta^d)$ is the Gamma density function. Finally, $p(\Psi)$ denotes the prior distribution over the model parameters.

With the likelihood function, the $Q$-function of the EM algorithm is defined by

$$Q = \mathbb{E}_{p(X_{0:K}, Y_{0:K}, \tau_{1:n_u}, \vec{m}_{1:n_u} \mid Z_{1:K}; \Omega^{(r)}, \Psi^{(r)})} \left[ \log P(Z_{1:K}, X_{0:K}, Y_{0:K}, \tau_{1:n_u}, \vec{m}_{1:n_u}; \Omega, \Psi) \right]$$

$$\approx \frac{1}{P} \sum_{p=1}^{P} \log P\left( Z_{1:K}, X_{0:K}^p, Y_{0:K}^p, \tau_{1:n_u^p}^p, \vec{m}_{1:n_u^p}^p; \Omega, \Psi \right) \tag{12}$$

Where $p\left( X_{0:K}, Y_{0:K}, \tau_{1:n_u}, \vec{m}_{1:n_u} \mid Z_{1:K}; \Omega^{(r)}, \Psi^{(r)} \right)$ is the posterior distribution of the latent variables given the current parameter estimates $\Omega^{(r)}$ and $\Psi^{(r)}$ at the $r$-th EM iteration.

The final line of Eq.12 provides an approximation of the $Q$-function using $P$ sample trajectories of the latent process. Algorithm 1 describes the steps used to construct these trajectories. Each trajectory may terminate with a different number of inducing points; the model explicitly accounts for this. At each time index, the procedure determines whether an inducing point should be removed from the trajectory or whether a new inducing point should be added. Consequently, at every time index the proposed SMC method produces a projected trajectory for future time points. Under the renewal formulation, the process retains the Markov property required for the recursive update of the inducing points.

## 3 RESULTS

In this section, we evaluate our framework on both simulated and real datasets. We begin with a one-dimensional chirp time series and then reconstruct a Lorenz system trajectory embedded in a higher-dimensional observation space, comparing MIP-CSDE performance against recent and established methods for time series analysis and manifold inference. Finally, we apply the model to neural recordings: decoding rat hippocampal place cell activity during navigation on a W-shaped maze Joo and Frank (2018) and inferring low-dimensional manifolds from monkey M1 and PMd activity during a center out reach task (Pei et al., 2021; Churchland et al., 2012a). Together, these analyses illustrate the key characteristics and advantages of our method.

### 3.1 CHIRP SIGNAL

We use MIP-CSDE to reconstruct a chirp signal whose frequency slowly decreases over time. Chirp signals provide a simple test of whether the model can place inducing points in ways that capture changing dynamics. We generated 500 noisy samples at 20 Hz, with frequency decreasing linearly from 0.2 to 0.1 Hz over 25 s. Both latent and observed processes were one-dimensional, and inducing point times were drawn from a Gamma prior that yields roughly one event every two seconds about 1.25 times the Nyquist rate for the highest chirp frequency.

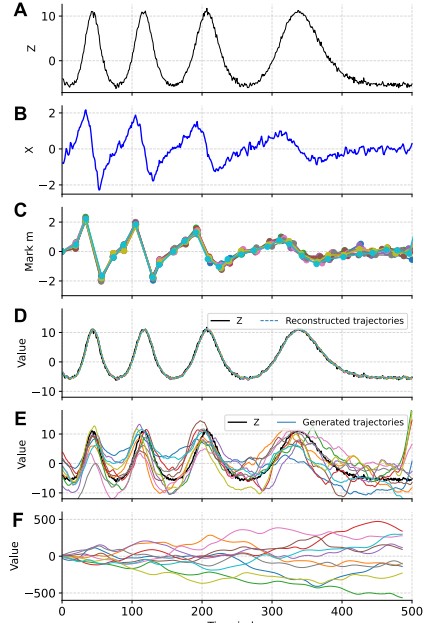

Figure 2: **Chirp Signal Reconstruction:** (A) Simulated noisy chirp signal. (B) Posterior mean of the latent process. (C) Inferred inducing events. (D) Reconstructed trajectory closely follows the observed data. (E-F) Samples generated from the trained and initial models. MIP-CSDE adapts event timing and values to track time varying frequency structure.

Inducing point values were drawn from a standard normal prior. We fit the model using our SMC-EM procedure with 1,000 particles over 12 iterations; smaller particle counts were also tested, and 1,000 provided stable fits across runs.

Figure 2 summarizes the results. The model accurately reconstructs the underlying trajectory and can generate new samples resembling chirp signals. Early in the sequence, the model increases inducing point magnitudes to track stronger oscillations; later, as the frequency slows, it uses more inducing points with smaller magnitudes. Unlike classical sampling approaches which typically reduce sample rates when frequency content decreases MIP-CSDE adapts through coordinated changes in both event timing and point values. These behaviors highlight the flexibility of the model and how its priors shape the resulting representation.

### 3.2 LORENZ DYNAMIC RECONSTRUCTION

We applied our framework to simulated data generated by the Lorenz attractor system (Lorenz, 1963), a standard test for models that aim to recover complex, nonlinear dynamics from noisy measurements. The true three-dimensional Lorenz trajectory was projected into a ten-dimensional observation space and sampled at 10 Hz for 500 time points, with added correlated Gaussian noise. Both latent processes, $X_t$ and $Y_t$, were set to three dimensions. We chose noise levels so that $X_t$ captured the rough dynamics while $Y_t$ remained smooth. During training, the model jointly learned the projection matrix $W$, the latent trajectory, and the timing and values of the inducing points.

Figure 1B shows the observed data and the inferred manifold. We used 20,000 particles and 25 EM iterations, which was sufficient for the estimates and likelihood to stabilize. The recovered trajectory closely follows the true Lorenz attractor, capturing both its overall shape and the finer curvature within each lobe. The inducing points adapt to these transitions: near the entrances and exits of each lobe, events become more frequent and their mark values change direction to anticipate the turn, whereas within a lobe the events become sparser and the marks remain smaller. This adaptive behavior is a core strength of MIP-CSDE it does not rely on uniform sampling or fixed basis functions, but instead adjusts where and how inducing events are placed, allowing it to reconstruct a highly nonlinear attractor from noisy, mixed observations.

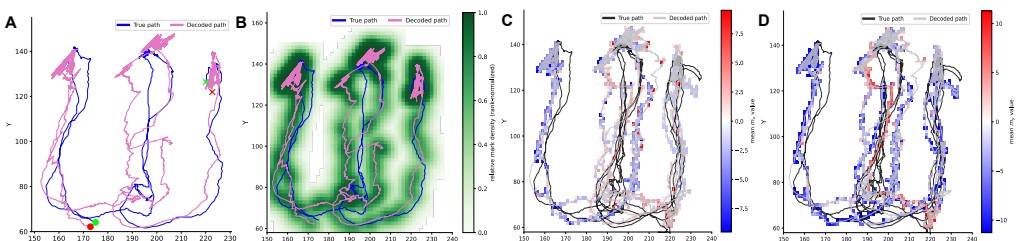

**Figure 3: MIP-CSDE place cell decoder on rat hippocampus data.** (A) True and decoded position for a left to right maze traversal. (B-D) Inducing events and their spatial marks concentrate near turns and rapid transitions, remaining sparse along straight segments. MIP-CSDE provides accurate and interpretable trajectory decoding from place cell spiking.

Table 1: Performance of MIP-CSDE compared to existing state of the art decoders. Root mean squared error (RMSE) quantifies the average distance between the decoded and true positions, and the 95% highest posterior density (HPD) coverage reports the fraction of time the true position lies inside the model's 95% credible region.

| Method | MIP-CSDE | GMM | Gaussian Approx | 4D GMM | Exact Point Process |
|---|---|---|---|---|---|
| RMSE, HPD% | 6.1, 92 | 17.2, 86 | 27.3, 62 | 15.7, 88 | 14.0, 91 |

To test whether the model recovers the correct dimensionality, we varied the latent dimension $d \in \{1, 2, 3, 4\}$ (Appendix Figure 7). Models with one or two dimensions collapsed the dynamics, and four dimensions introduced unnecessary freedom without improving accuracy. The three-dimensional model matching the true system gave the best reconstruction and prediction, confirming that MIP-CSDE identifies an appropriate latent dimensionality for this dataset.

### 3.3 RAT HIPPOCAMPUS: DECODING SPATIAL TRAJECTORIES FROM CA1 SPIKING

We applied MIP-CSDE to hippocampal CA1 place cell recordings from a rat running on a W-shaped maze Joo and Frank (2018). Decoding position from place cell activity is a standard benchmark, with prior work using Gaussian mixture models and deep neural network based decoders to improve robustness and accuracy Yousefi et al. (2019); Karlsson and Frank (2008); Brown et al. (1998). The dataset contains spiking activity from 62 cells sampled in 33 ms bins. We modeled the latents $X_k$ and $Y_k$ as two-dimensional, where $X_k$ captures movement related dynamics and $Y_k$ represents position. Each cells activity in bin $k$ is $Z_k = \{z_k^i\}_{i=1}^{62}$, where $z_k^i \in \{0, 1\}$. Firing rates $\lambda_i$ were estimated using standard kernel methods Yousefi et al. (2019), and the likelihood for each cell followed a Poisson point process model:

$$p(Y_k \mid z_k^i) \propto p(z_k^i \mid \lambda_{i,k}) = (\Delta k \, \lambda_{i,k})^{z_k^i} \exp(-\lambda_{i,k} \Delta k) \tag{13}$$

with $\Delta k = 33$ ms. The full likelihood is the product across cells. We used the first 80% of the trajectory to fit rate models and decoded position on the remaining 20%, following common practice Yousefi et al. (2019). Decoding used a particle filter with 10,000 particles, which is consistent with earlier hippocampal decoders Yousefi et al. (2019); Brown et al. (1998) and provided stable posterior trajectories. For the inducing point process, we used a Gamma$(8, 2)$ prior for waiting times (mean 4 s) and a Gaussian prior $\mathcal{N}([0, 0]^\top, 0.5 I_{2 \times 2})$ for mark values. These choices mildly favor several second gaps between events and prevent unrealistically large mark jumps, while remaining flexible; moderate changes to the hyperparameters did not affect the results.

Figure 3 shows the decoded trajectory and inducing points. The posterior mean path closely follows the rat's movement along the maze arms (panel A). Inducing events cluster near corners and the central corridor (panel B), where the trajectory changes direction more rapidly, and remain sparse along straight segments. Additional diagnostics in Appendix Figure 8 show stable latent dynamics across the full session. Table 1 compares MIP-CSDE with existing decoders: it achieves lower RMSE and comparable or higher 95% HPD coverage. With a runtime of about 2 ms per 33 ms bin, MIP-CSDE provides an accurate, temporally aligned, and interpretable decoder for hippocampal place cell activity.

### 3.4 MANIFOLD DIFFERENTIATION DURING MONKEY REACHING TASK

We applied MIP-CSDE to the Neural Latents Benchmark (NLB) MC_Maze dataset (Pei et al., 2021), which contains high resolution recordings from macaque PMd and M1 during a center out reaching task (Churchland et al., 2012a). The dataset includes spiking activity from 182 neurons sampled at 1 ms, along with hand position, velocity, and cursor information. Prior work shows that population activity at movement onset strongly predicts the upcoming reach (Churchland et al., 2012a; Pei et al., 2021).

Our goal was to learn a low-dimensional representation of neural activity during preparation and movement, and to characterize how these phases relate. Building on studies suggesting that preparatory activity sets the initial conditions for movement on a shared manifold (Churchland et al., 2012a; Kaufman et al., 2014; Elsayed et al., 2016), we tested whether preparatory and reach trajectories are linked by a simple linear transformation. For interpretability, we restricted MIP-CSDE to a two-dimensional latent space (though the model can accommodate higher dimensions). Each neurons spiking was modeled with a Bernoulli GLM, where the log odds of spiking at time $k$ is a linear function of the latent state, allowing the latents to be interpreted as low-dimensional predictors of firing probability.

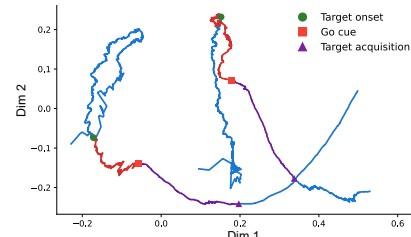

Figure 4: **Inferred neural manifolds during preparation and reach.** Two-dimensional latent trajectories from two MC_Maze trials. For each trial, preparatory and reach activity lie in the same 2D space and form smooth, connected paths, while trajectories differ across reach targets. A single affine map fit between preparation and movement explains about 90% of the variance in the reach phase, showing that movement activity largely reflects a linear transformation of preparatory activity within this shared manifold.

Figure 4 shows the inferred 2D manifolds for two example trials. Within each trial, preparatory and reach activity occupy overlapping regions of the same latent space and form smooth, connected trajectories, while across trials the paths diverge according to the reach target. To quantify the link between preparation and movement, we aligned the two trajectories and fit a single affine map $Y(t) \approx AX(t) + b$. This $2 \times 2$ transform explained about 90% of the variance in the reach phase latents ($R^2 \approx 0.90$), showing that movement activity is well approximated as a linear transformation of preparatory activity within the same 2D manifold. This finding aligns with prior MC_Maze results and supports the view that preparatory dynamics organize the neural state, which is then smoothly transformed into the reach trajectory.

## 3.5 Comparative Analysis on Simulated Datasets

To evaluate the performance of MIP-CSDE, we benchmark it against several continuous time baselines in terms of both predictive accuracy and computational efficiency. The baselines include models of increasing flexibility: a Linear SDE, a Gaussian Process SDE (GP-SDE) (Duncker et al., 2019), and the Gaussian Process Switching Linear Dynamical System (GP-SLDS) (Hu et al., 2024). To ensure a robust comparison, all baseline models were fitted using the highly efficient SING inference framework (Hu et al., 2025).

We report performance using the MSE in the observation space. This choice of metric is necessitated by a key property of the baseline models, where their latent spaces are only identified up to an arbitrary affine transformation. This implies that a direct comparison between their inferred latent trajectories and the ground truth is not meaningful without a post hoc alignment procedure. The reconstructed latent along with realigned version is presented in appendix A.6. The quantitative results

Table 2: Performance comparison with baseline continuous time models. For simulated data (Chirp, Lorenz), we report MSE. (Results are reported as mean $\pm$ standard error across 5 trials.)

| Dataset | MIP-CSDE | Linear SDE | GP-SDE | GP-SLDS |
|---|---|---|---|---|
| Chirp Signal | $0.30 \pm 0.02$ | $0.48 \pm 0.05$ | $0.39 \pm 0.06$ | $0.36 \pm 0.02$ |
| Lorenz System | $0.18 \pm 0.01$ | $0.28 \pm 0.04$ | $0.25 \pm 0.03$ | $0.21 \pm 0.03$ |

in Table 2 indicate that MIP-CSDE attains the lowest error on the Lorenz system (MSE $0.18 \pm 0.01$), while performing comparably to GP-SDE on the chirp signal (MSE $0.30 \pm 0.02$ vs. $0.29 \pm 0.06$). On the Lorenz benchmark, our implementation required approximately $290\,\mathrm{s}$, compared with $12\,\mathrm{s}$ for Linear SDE, $38\,\mathrm{s}$ for GP-SDE, and $252\,\mathrm{s}$ for GP-SLDS. We also observed higher memory usage for GP-SLDS on longer sequences, which is consistent with the scaling behavior of GP-based kernels. For MIP-CSDE, runtime scales approximately linearly with the number of particles increasing particles improves accuracy at additional computational cost, reflecting an explicit accuracy compute trade off. Although implementation details and hardware choices affect absolute timings, the ob-

served trends are consistent with the computational analysis presented in Section 2.3.2. As expected, measured runtimes scale approximately linearly with both particle count and sequence length (Appendix A.7). Overall, these results support MIP-CSDE as a more accurate and computationally competitive approach for recovering latent dynamics and manifolds in continuous time.

## 4    DISCUSSION

We developed a training and inference pipeline for a discretized version of our framework and evaluated it on both simulated and neural datasets. Across these examples, MIP-CSDE consistently adapted its inducing events to capture the structure of the underlying dynamics. In the chirp signal, the model allocated events in response to changes in frequency; in the Lorenz system, it recovered the nonlinear geometry of a chaotic attractor and scaled effectively as dimensionality increased. In the rat hippocampus task, the model served as a robust decoder of spatial position, and in the monkey reaching task it revealed a shared manifold linking preparatory and movement activity through a simple linear transformation. Together, these findings show that the model is flexible enough to handle diverse forms of temporal structure and strong nonlinearities while remaining computationally efficient. We also compared MIP-CSDE to continuous time baselines including Linear SDE, GP-SDE, and GP-SLDS. On simulated datasets, MIP-CSDE matched or exceeded their reconstruction accuracy while retaining linear scaling in data length. These comparisons highlight the advantages of combining adaptive inducing events with a cascaded SDE formulation: the model avoids the cubic scaling of GP based approaches while still achieving competitive performance, even on systems with complex latent geometry. The simulated results further support the universal approximation properties of the framework, as the model accurately reconstructed both smooth and chaotic dynamics.

While models such as DKF and GP based methods (Krishnan et al., 2015b; Casale et al., 2018) can also infer latent processes, they often depend on large datasets, specific kernel choices, or more complex inference procedures. MIP-CSDE requires no pre-defined kernel and offers two complementary modes of interpretation: one can examine the latent trajectories directly or study the inducing point timings and values that drive them. This dual interpretability is valuable in neuroscience, where linking latent dynamics to behavior or external variables is often a key goal (Churchland et al., 2012b).

Although we used a discretized version of the model, a continuous formulation similar to neural ODEs (Chen et al., 2018) could further improve inference stability and reduce numerical error when generating trajectories. Several parameters, such as the noise variance in $X_t$ and the decay term in $Y_t$, influence the trade off between smoothness and flexibility, and learning them automatically rather than setting them manually is a natural next step. Our inference method is based on a custom SMC algorithm that handles both Gaussian and point process observations. In settings where the latent to observation mapping is linear and Gaussian, the model reduces to a standard linear state space system, suggesting a hybrid approach in which SMC infers inducing events while the Kalman filter updates states, offering additional computational gains.

Finally, we assumed independence between waiting times and marks, which simplifies inference but may limit adaptability across temporal scales. More expressive choices such as time dependent or history dependent distributions, or joint modeling of marks and waiting times may allow the model to better handle rapid transitions. The model also requires choosing a latent dimensionality. Although we examined several choices in the Lorenz example, a principled selection method such as automatic relevance determination (ARD) Wipf and Nagarajan (2007) may enable the model to infer the effective latent dimension directly from data.

## 5    CONCLUSION

Here, we introduce a cascade SDE framework to infer the underlying latent structure and manifold present in high-dimensional time series using a sparse set of inducing points, adaptively placed in both time and value space. The model achieves a high level of expressive power, while its computational cost grows only linearly in both data dimensions and time. The comparative analyses indicate that it achieves performance accuracy on par with or superior to state-of-the-art models. These results suggest that the model holds promise as an unsupervised dimensionality reduction tool and can be robustly applied as a dynamical neural decoder or adaptive feature extractor across a range of neuroscience applications and time-series analysis.

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

# A APPENDIX

## A.1 NONPARAMETRIC AND NON-MARKOVIAN PROPERTIES

In our model, the $X_t$ exhibits dynamics similar to that of samples from a GP with specific covariance structures, for example, an Ornstein-Uhlenbeck process corresponding to an exponential kernel (Uhlenbeck and Ornstein, 1930). Within the model, the number of inducing points is not fixed in advance and it adapts dynamically to the complexity of the observed dynamics. The nonparametric and GP-like nature of our model makes it an alternative choice for machine learning and probabilistic applications. In our proposed model, the trajectory of $X_t$ process is dependent to both past and future event-value pairs; thus it does not satisfy the Markovian property. This might complicate both training and inference within our framework. In section 2.2 , we reformulated the inducing point distribution using a renewal marked point process, which lets us to define $X_t$ and $Y_t$ process with a Markovian property. In section 2.4, we leverage this reformulation of SDEs for the inference and training of the model.

## A.2 CONVERGENCE OF SPACING BETWEEN ADJACENT SAMPLES

Let $X_1, X_2, \ldots, X_N$ be i.i.d. random variables uniformly distributed on $[0, T]$, and let $X_{(1)} \leq X_{(2)} \leq \cdots \leq X_{(N)}$ denote their order statistics. Define the adjacent spacings $d_i = X_{(i+1)} - X_{(i)}$ for $i = 1, \ldots, N-1$.

From properties of uniform order statistics, the expected value of the $i$-th order statistic is $\mathbb{E}[X_{(i)}] = \frac{iT}{N+1}$. It follows that

$$\mathbb{E}[d_i] = \mathbb{E}[X_{(i+1)} - X_{(i)}] = \frac{T}{N+1} \tag{14}$$

which satisfies $\mathbb{E}[d_i] \to \frac{T}{N}$ as $N \to \infty$.

Moreover, the variance of $d_i$ satisfies $\mathrm{Var}(d_i) = O(N^{-2})$. By Chebyshevs inequality,

$$\mathbb{P}\left(|d_i - \mathbb{E}[d_i]| \geq \epsilon\right) \leq \frac{\mathrm{Var}(d_i)}{\epsilon^2} = O\left(\frac{1}{N^2}\right) \tag{15}$$

which vanishes as $N \to \infty$. Therefore, $d_i \to \frac{T}{N}$ in probability.

Furthermore, classical results on uniform spacings imply that the maximum spacing $\max_i d_i$ satisfies

$$\max_{1 \leq i \leq N-1} d_i = \frac{T}{N} + O\left(N^{-1/2}\right) \tag{16}$$

with high probability, confirming that all gaps become uniformly close to $\frac{T}{N}$ as $N \to \infty$.

## A.3 VARIATIONAL INFERENCE FOR MODEL TRAINING

To complement the EM approach, we also develop a variational inference (VI) algorithm for training our model. As in Section 2.2, we begin with the discrete-time representation of the renewal process, which breaks the non-Markovian dependence of $X_t$ by ensuring that the next inducing point and its mark are known at any time $t$. This property makes recursive inference feasible, but instead of relying on exact latent sampling as in the E-step of EM, we approximate the intractable posterior using a variational family.

Specifically, we introduce the following structured mean-field approximation:

$$q_\phi\left(X_{0:K}, Y_{0:K}, \{\tau_i, m_i\}_{i=1}^L\right) = q_X(X_{0:K}; \phi_X)\, q_Y(Y_{0:K}; \phi_Y)\, q_{\tau,m}(\{\tau_i, m_i\}_{i=1}^L; \phi_{\tau,m}), \tag{17}$$

where $\phi = \{\phi_X, \phi_Y, \phi_{\tau,m}\}$ are variational parameters. This factorization decouples states and inducing points while retaining the renewal structure. In practice, we amortize these distributions using neural networks that map observed data into variational parameters.

The training objective is the evidence lower bound (ELBO):

$$\mathcal{L}(\theta, \phi) = \mathbb{E}_{q_\phi}\Big[\log p_\theta(Z_{0:K}, X_{0:K}, Y_{0:K}, \{\tau_i, m_i\}_{i=1}^L) - \log q_\phi(X_{0:K}, Y_{0:K}, \{\tau_i, m_i\}_{i=1}^L)\Big], \quad (18)$$

where $\theta$ denotes the generative model parameters. Maximizing $\mathcal{L}(\theta, \phi)$ yields both approximate posterior inference (via $q_\phi$) and maximum likelihood estimation of $\theta$.

We employ stochastic gradient variational Bayes (SGVB) with the reparameterization trick to obtain low-variance gradient estimates. In this formulation, sampling of event–value pairs is embedded directly into the variational distribution $q_{\tau,m}$, which is parameterized by waiting-time and mark distributions. These distributions can be chosen flexibly, e.g., Gamma and Gaussian, or replaced with neural flows for greater expressivity.

---

**Algorithm 2** Variational Inference for Inducing Point and State Estimation

---

1: **Initialize:** model parameters $\theta$, variational parameters $\phi$
2: **for** each training iteration **do**
3:     Sample latent variables from $q_\phi$:

$$X_{0:K}, Y_{0:K}, \{\tau_i, m_i\}_{i=1}^L \sim q_\phi$$

4:     Compute stochastic ELBO estimate:

$$\widehat{\mathcal{L}} = \log p_\theta(Z_{0:K}, X_{0:K}, Y_{0:K}, \{\tau_i, m_i\}_{i=1}^L) - \log q_\phi(X_{0:K}, Y_{0:K}, \{\tau_i, m_i\}_{i=1}^L)$$

5:     Update $(\theta, \phi)$ via gradient ascent on $\widehat{\mathcal{L}}$
6: **end for**

---

Unlike EM, where process noise parameters are typically fixed, VI allows them to be included in the variational family and learned directly. In practice, however, we sometimes constrain these parameters to preserve stability of the underlying SDEs.

Finally, while the description above applies to a single observed trajectory, the variational framework naturally extends to multiple trials. Each trial maintains its own approximate posterior over inducing points and latent trajectories, while global parameters $\theta$ are shared across trials. This amortized formulation enables scalable training across large experimental datasets.

A.4    DISCRETE REPRESENTATION OF HIERARCHICAL SDE

We focus on a discrete-time formulation of the model. To construct the discrete process, we assume that $X_t$ and $Y_t$ are sampled at regular intervals of $\Delta t$. The discrete representation of $x_t^d$ is defined as:

$$x_{k+1}^d = x_k^d + \frac{m_{i+1}^d - x_k^d}{t_{i+1} - k\Delta t}\Delta t + \sqrt{\frac{(t_{i+1} - k\Delta t)(k\Delta t - t_i)}{t_{i+1} - t_i}\Delta t} \cdot w_k^d,$$
$$w_k^d \sim \mathcal{N}(0, \sigma_x^{d^2}) \tag{19}$$

where $w_k^d$ is a Gaussian noise term. Similarly, the discrete-time evolution of $y_t^d$ is:

$$y_{k+1}^d = y_k^d + x_k^d \Delta t + \sqrt{\Delta t} \cdot \nu_k^d, \quad \nu_k^d \sim \mathcal{N}(0, \sigma_y^{d^2}) \tag{20}$$

where $\nu_k^d$ is Gaussian noise. The discrete observation process is given by:

$$Z_k = WY_k + \xi_k, \quad \xi_k \sim \mathcal{N}(0, R) \tag{21}$$

where $Y_k = \begin{pmatrix} y_k^1 \\ \vdots \\ y_k^D \end{pmatrix}$, $W \in \mathbb{R}^{M \times D}$ is a projection matrix, and $\xi_k$ is observation noise.

To ensure accuracy, $\Delta t$ must be much smaller than the minimum inter-event time, i.e., $\Delta t \ll \min(\tau_i)$, so that no two inducing points fall within the same discrete time bin. This constraint can be satisfied by analyzing the posterior distribution of waiting times and adjusting $\Delta t$ accordingly. In essence, we require an orderly event process—allowing at most one event per bin—which can be enforced by carefully selecting the bin size $\Delta t$.

When using the waiting time representation, at time $k$, we already know when the next inducing point (e.g., event 213) will occur and what its value will be. From a Markovian perspective, we can assume that the entire process is determined at time $k$. This assumption simplifies the inference procedure presented in Algorithm 1.

### A.5 GAMMA DISTRIBUTION FOR WAITING TIMES AND PRIOR SELECTION FOR INDUCING POINTS

To complete the Bayesian framework, we define priors for the model parameters. For the mark distribution parameters, we assume:

$$\mu_i \mid \Sigma_i \sim N(\mu_0, \xi\Sigma_i), \quad \Sigma_i \sim \text{Inverse-Wishart}(\nu, \Psi) \tag{22}$$

where $\mu_0, \xi, \nu$, and $\Psi$ are hyperparameters. For the Gamma distribution parameters $\alpha$ and $\xi$ governing the waiting times $\tau_i$, we consider the following prior options based on domain-specific knowledge, though their specific forms remain to be fully specified in this study:

- For $\alpha$:
  - $\alpha \sim \text{Gamma}(a_0, b_0) = \frac{b_0^{a_0}}{\Gamma(a_0)} \alpha^{a_0-1} e^{-b_0 \alpha}$,
  - $\alpha \sim \text{Exp}(\xi_0) = \xi_0 e^{-\xi_0 \alpha}$,
  - $\alpha \sim \text{Lognormal}(\mu_0, \sigma_0^2) = \frac{1}{\alpha\sqrt{2\pi\sigma_0^2}} \exp\left(-\frac{(\log \alpha - \mu_0)^2}{2\sigma_0^2}\right)$,

- For $\xi$:
  - $\xi \sim \text{Gamma}(c_0, d_0) = \frac{d_0^{c_0}}{\Gamma(c_0)} \xi^{c_0-1} e^{-d_0 \xi}$,
  - $\xi \sim \text{InvGamma}(\gamma_0, \delta_0) = \frac{\delta_0^{\gamma_0}}{\Gamma(\gamma_0)} \xi^{-(\gamma_0+1)} e^{-\delta_0/\xi}$,

where $a_0, b_0, \xi_0, \mu_0, \sigma_0^2, c_0, d_0, \gamma_0, \delta_0$ are hyperparameters.

In our model, the waiting times $\tau_i$ and the marks $\vec{m}_i$ associated with each event are generated according to specific probabilistic distributions:

- **Waiting Times $\tau_i$:**
  The waiting times between events are assumed to follow a Gamma distribution parameterized by a shape parameter $\alpha$ and a rate parameter $\lambda$. The probability density function for $\tau_i$ is given by:
  $$p(\tau_i) = \text{Gamma}(\tau_i; \alpha, \beta) \tag{23}$$
  where the Gamma distribution is defined as:
  $$\text{Gamma}(\tau_i; \alpha, \beta) = \frac{\beta^\alpha}{\Gamma(\alpha)} \tau_i^{\alpha-1} e^{-\beta\tau_i}, \quad \tau_i > 0 \tag{24}$$
  and $\Gamma(\alpha)$ denotes the Gamma function evaluated at $\alpha$.

  **Motivation for using the Gamma distribution:**
  Consider $N$ i.i.d. samples $U_1, \ldots, U_N \sim \text{Uniform}(0, T)$, and denote their order statistics by $U_{(1)} \leq \cdots \leq U_{(N)}$. Define the gaps between consecutive order statistics as
  $$\Delta_0 = U_{(1)}, \quad \Delta_i = U_{(i+1)} - U_{(i)} \text{ for } i = 1, \ldots, N-1, \quad \Delta_N = T - U_{(N)} \tag{14}$$

As $N \to \infty$, it is well-known that each gap satisfies $\Delta_i \xrightarrow{p} T/N$, and the rescaled gaps $N\Delta_i$ converge in distribution to an exponential random variable, that is,

$$N\Delta_i \xrightarrow{d} \text{Exp}(1) \tag{15}$$

Moreover, the normalized gaps $(\Delta_0/T, \ldots, \Delta_N/T)$ jointly follow a Dirichlet$(1, \ldots, 1)$ distribution. Marginally, each normalized gap $\Delta_i/T$ follows a Beta$(1, N)$ distribution. As $N$ becomes large, the Beta$(1, N)$ distribution approximates a Gamma$(1, 1/N)$ distribution, because

$$N \cdot (\Delta_i/T) \xrightarrow{d} \text{Exp}(1) \tag{16}$$

which suggests that

$$\Delta_i \approx \text{Gamma}(1, T/N) \tag{17}$$

Thus, in the large-sample limit, the gaps between ordered uniform samples behave approximately like scaled exponential random variables.

To simulate ordered points efficiently for a finite number $M$ of samples, we propose sampling $M$ independent gaps

$$\Delta_i \sim \text{Gamma}(1, T/M) \tag{18}$$

and constructing ordered points via the cumulative sums

$$U_{(i)} = \sum_{j=0}^{i-1} \Delta_j, \quad i = 1, \ldots, M \tag{19}$$

This motivates our use of Gamma-distributed waiting times $\tau_i$ in the model, capturing the natural variability in the timing of events.

- **Marks $\vec{m}_i$:**
  The marks, representing additional information associated with each event, are modeled as drawn from a multivariate normal (Gaussian) distribution. Each mark vector $\vec{m}_i$ has an associated mean vector $\mu_i$ and covariance matrix $\Sigma_i$, with the distribution:

$$p(\vec{m}_i) = \mathcal{N}(\vec{m}_i; \mu_i, \Sigma_i) \tag{25}$$

explicitly given by:

$$\mathcal{N}(\vec{m}_i; \mu_i, \Sigma_i) = \frac{1}{(2\pi)^{d/2} |\Sigma_i|^{1/2}} \exp\left( -\frac{1}{2} (\vec{m}_i - \mu_i)^\top \Sigma_i^{-1} (\vec{m}_i - \mu_i) \right) \tag{26}$$

where $d$ is the dimensionality of the mark vector.

This modeling choice allows flexible and realistic characterization of the temporal dynamics $\tau_i$ and the event-related features $\vec{m}_i$ within the system under study.

A.6 COMPARISON OF LATENT TRAJECTORIES

In this section, we provide additional analyses of the latent trajectories inferred by the compared models. As discussed in the main text, the latent spaces of baseline continuous-time models (e.g., Linear SDE, GP-SDE, GP-SLDS) are identifiable only up to an arbitrary affine transformation. To enable meaningful comparisons, we apply a Procrustes-based alignment procedure between the inferred latent trajectories and the ground-truth latent dynamics.

Representative examples are shown in Figure 5, where we compare raw latent trajectories (top panels) with their aligned counterparts (bottom panels) across baseline models. This visualization highlights the necessity of alignment for baseline approaches, as their raw latents are not directly comparable to the true dynamics.

The generative structure of MIP-CSDE naturally constrains its latent space, yielding trajectories that are more directly interpretable without alignment. Nonetheless, for fairness, all quantitative performance metrics reported in the main text are computed in the observation space.

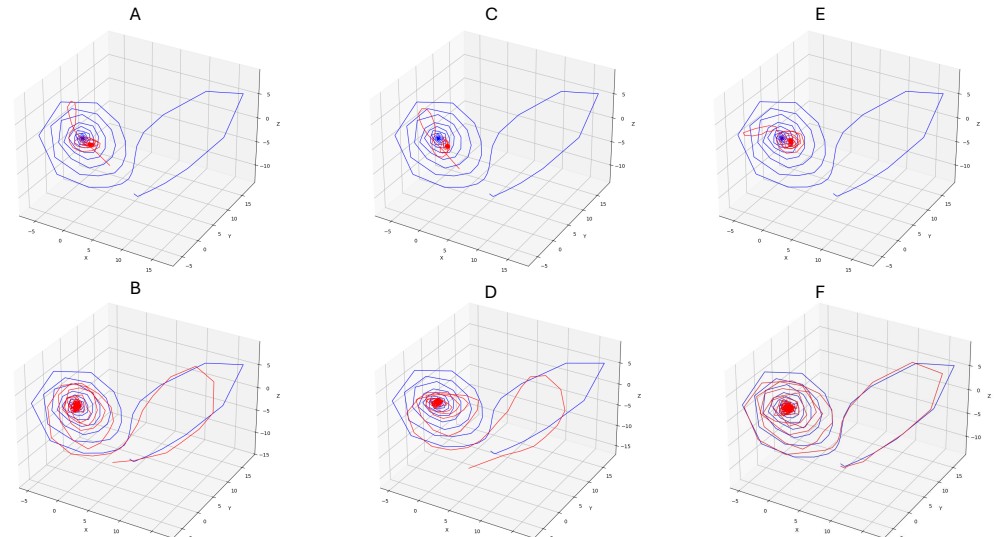

Figure 5: **Reconstruction of the Lorenz Trajectory Using Baseline Models.** Panels (A–B): Linear SDE; (C–D): GP-SDE; (E–F): GP-SLDS. For each model, the top panel shows the raw reconstructed latent trajectory, and the bottom panel shows the trajectory after Procrustes alignment to the ground truth.

## A.7 RUNTIME SCALABILITY EXPERIMENTS

This appendix examines the empirical runtime scaling of the SMC–EM procedure used in our experiments. We vary two primary factors that drive computational cost: the number of particles $P$ in the SMC layer and the sequence length $K$ (number of time bins). For each setting, we run the Lorenz benchmark ten times with independent random seeds and report wall-clock time averaged over runs. Following common practice for GPU timing, we insert explicit CUDA synchronizations around the timed region and use a high-resolution host timer; we discard a short warm-up to avoid one-time kernel compilation and cache effects. All experiments use the same model configuration and batch size as in the main results to isolate the effect of $P$ and $K$.

Figure 6 summarizes the measurements. Panel (A) varies $P$ at fixed $K$, plotting seconds per 1,000 time bins on a log scale. Panel (B) varies $K$ at fixed $P = 20{,}000$, reporting seconds per 20,000 particles. In both regimes, ordinary least squares fits (orange) achieve $R^2 \geq 0.995$ against the measured times (blue), consistent with the expected $\mathcal{O}(P)$ and $\mathcal{O}(K)$ complexity under our implementation. Absolute times depend on hardware, kernel fusion, and memory bandwidth, but the trends align with the cost analysis in Section 2.3.2. We note that memory usage grows linearly in $P$ and modestly in $K$ due to buffering of particle states; for large $P$, gradient checkpointing and mixed precision can reduce footprint without materially affecting the observed scaling.

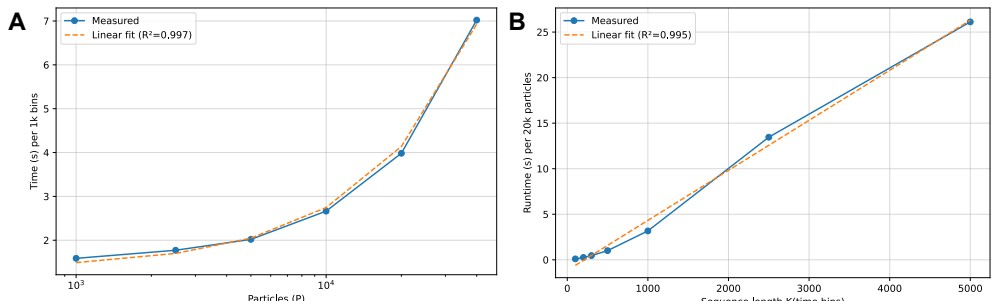

Figure 6: **Runtime of the SMC–EM Algorithm With Varying Numbers of Particles on the Lorenz Benchmark.** (A) Runtime vs. particle count $P$, shown on a log scale (seconds per 1k bins).(B) Runtime vs. sequence length $K$ (seconds per 20k particles). Results are averaged over 10 runs. In both cases, measured runtime (blue) closely follow linear fits (orange, $R^2 \geq 0.995$), confirming the expected $O(P \cdot N)$ complexity discussed in Section 2.2.2. Experiments were run on an NVIDIA T4 GPU (16 GB) using PyTorch + CUDA.

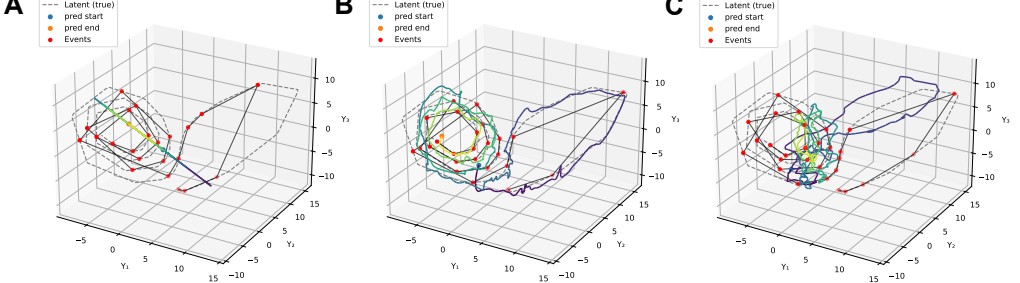

Figure 7: **Effect of latent dimensionality on MIP-CSDE decoding for the Lorenz system.** (A) One-dimensional latent model: decoded trajectory for a single observed coordinate of the Lorenz system, illustrating underfitting when the latent space is too restrictive. (B) Two-dimensional latent model: decoded trajectory with improved reconstruction of the underlying Lorenz dynamics compared to the 1D case. (C) Four-dimensional latent model: decoded trajectory capturing finer structure of the Lorenz dynamics, showing that increasing latent dimensionality allows MIP-CSDE to represent more of the systems nonlinear behavior.

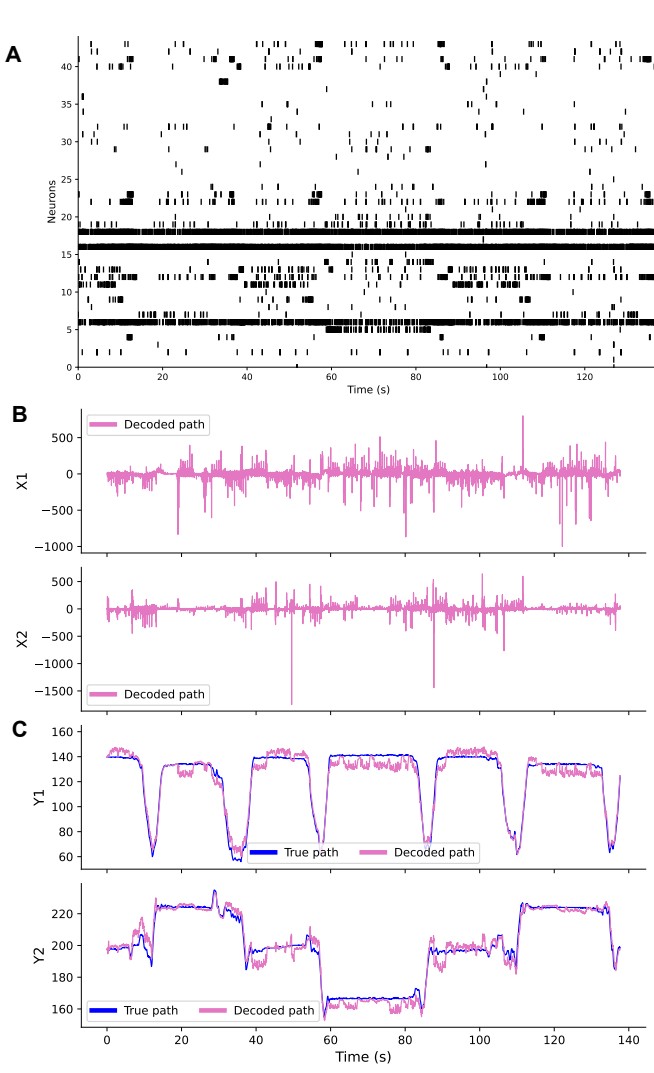

Figure 8: **MIP-CSDE decoding of 2D position from hippocampal population activity.** (A) Raster plot of spiking activity from 62 simultaneously recorded hippocampal neurons during a representative segment of the rats movement. Each row corresponds to one neuron and each tick marks a spike. (B) Inferred mean velocity along the horizontal and vertical axes over time, obtained from the MIP-CSDE latent state and mapped into velocity coordinates. The model captures the main changes in movement speed and direction. (C) Decoded mean 2D position compared with the rats true trajectory for the same segment. The close overlap between decoded and true paths illustrates that MIP-CSDE can accurately reconstruct position from the observed spike trains.

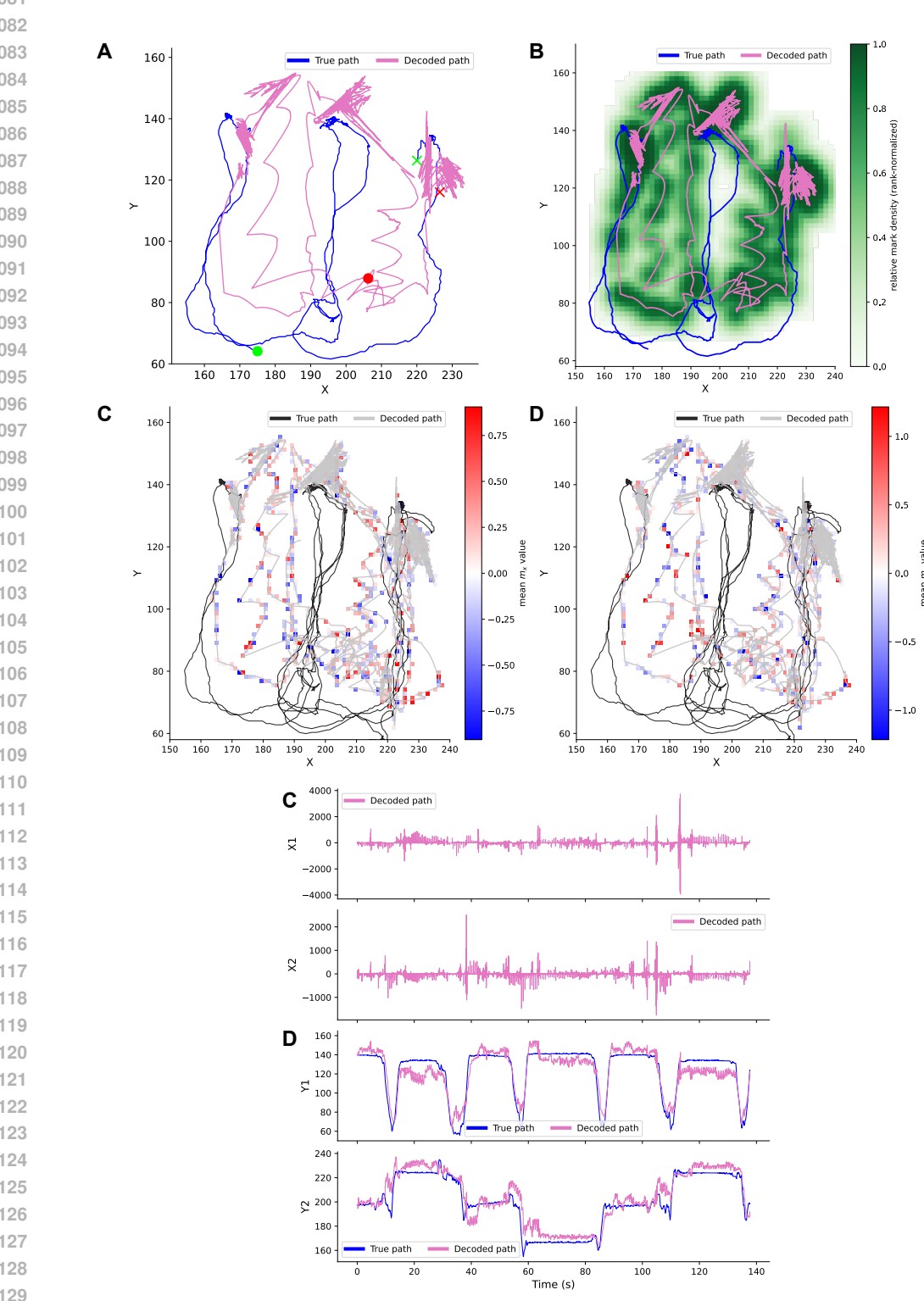

Figure 9: **MIP-CSDE rat decoding with 1,000 particles.** (A) One example traversal of the W-maze, showing the rat moving from the left arm to the right arm. The model is trained and decoded on the full recording session, but for visual clarity we display a single left-to-right trajectory. (B) Inducing-point locations in latent vs. observed position for this traversal. (CD) Heatmaps of inferred $x$- and $y$-coordinate marks along the same latent trajectory. (E) Inferred mean horizontal and vertical velocity over time. (F) Decoded mean 2D position vs. true trajectory, showing accurate reconstruction with 1,000 particles.

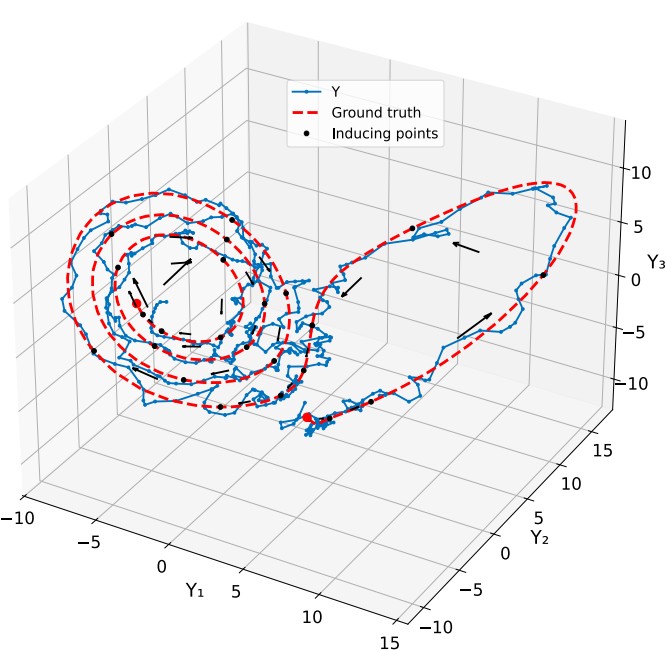

Figure 10: **Decoding of Lorenz Trajectory Using MIP-CSDE.** Here, the projection matrix from the latent process to observations and the additive noise covariance in the simulated data are known. The plot shows one inferred trajectory ($Y$), and dots indicate a sample set of inducing points. The decoded trajectory is aligned with the generated trajectory.

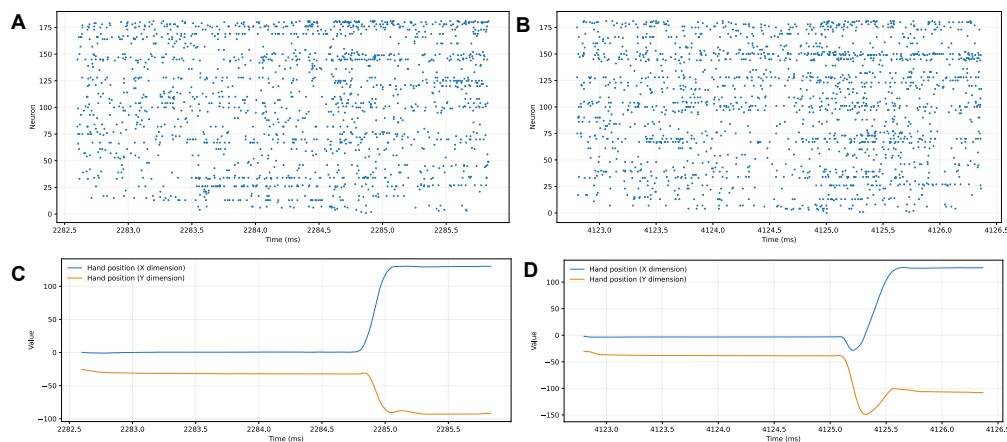

Figure 11: **Raster Plots of Monkey PDM and M1 Neurons Along with Hand Position During the Reach Task.** (A, B) Raster plots of 182 neurons from M1 and PMd, showing activity before target onset, the go cue, and target acquisition across two task trials. (C, D) Corresponding monkey hand positions during the same trials.

