# OpenReview forum: "MARKED INDUCING POINT CASCADED SDES FOR NEURAL MANIFOLD LEARNING"
_ICLR.cc/2026/Conference — ICLR 2026 Conference Desk Rejected Submission_

### Official Review · Reviewer_z9Qw · 2025-10-27

**Soundness:** 2
**Presentation:** 2
**Contribution:** 3
**Rating:** 2
**Confidence:** 3

**Summary:**

Assuming that many high-dimensional time series are governed by simpler dynamics on a low-dimensional manifold, this work presents a time series model designed to extract this low-dimensional structure, advertised as a sweet spot between interpretability and expressive power. Specifically, the model uses a cascade SDE model that is fit to data using expectation maximation (or alternatively variational inference).
Several theoretical properties of the model are studied, i.e., universal approximation capabilities and computational costs.
Empirically, the model is evaluated in 4 experiments or tasks, (i) reconstruction of a chirp signal, (ii) a Lorenz system, (iii) decoding animal spacial trajectories from brain data, and (iv) inference of a manifold representation of neural activity on the MC_Maze neural latents benchmark dataset.

**Strengths:**

- The expressivity of the presented method is studied theoretically and a universal approximation theorem is proven.
- The method scales with O(sampled time points (N) times sampled trajectories (P)), whereas competitors can scale with O(N$^3$)
- Empirically, the model is able to reconstruct the studied signals, slightly outperforming the baselines.

**Weaknesses:**

- **Clarity.** It was quite difficult for me to understand the descriptions and arguments in the article, and I don't think I was successful.
 This includes the proofs and setting of the theorems, the model-fitting procedure and the setting and outcome of the experiments. More specifically, ...
    - Experiments. The overall setting of the experiments is insufficiently described, and as a result I am still not sure of what the goal was. From my understanding (but I might be completely wrong) it seems that the input to the model is a (very finely) sampled time series of latent states $X_t$ and that the model output is a sample path of a an estimated stochastic process $\hat Z_t$ that is compared to ground truth observations $z_t$. It also seems that the model is fit on a single time series (instead of a corpus of time series) and that only one trajectory pair $(x_t, z_t)$ is available for model fitting, i.e. spread information to fit the stochastic process is not directly accessible.
    - In the proof of the Theorem, the construction of the $x_t^{d,N}$ and their relation to the marks $m_i$ is unclear. Therefore, also Eq. 8 (and arguments build upon it) is unclear.
    - It seems that the proof of the theorem assumes that the time series can be observed arbitrarily often and at equally spaced time points, but this is not stated as an assumption.
    - Model fitting, it is stated that model fitting *"involves updating several sets of parameters"*, it would be very beneficial to the reader to specify the model parameters and Likelihood function here.

- The model is advertised for extracting low dimensional representations of time series, suggesting that obtaining high quality representations is the goal and strength of the model. However, the evaluation of the obtained representations is quite limited. Only on one dataset (experiment 3.5),  insights are obtained from the representations.
Specifically, the authors observe that  *"within each trial, the trajectories during the reach and preparation phases were similar, but they differed across trial"*. However, what is meant with *"similar"* is not specified, so to me it remains unclear, in which way they are similar or how I can see it in the plots (Figure 4 also misses a legend entry for the blue trajectories).
- I might have misunderstood the experiment design, but the tasks seem quite easy. Reconstructing the chirp signal is based on 500 samples and the added gaussian noise is quite small (variance 0.1). Still, the model is unable to reconstruct the signal near its peaks.
For the rat-hippocampus data, we even have 30 observations per second for more than 120 seconds and use 80% of that for model fitting. If only reconstruction is the goal, then, given the amount of available data, this should be quite easy. While one might argue that constructing the rat movement only from the spiking activity of cortex cells is more difficult (if this is what is done in experiment 3.4), it should be noted that the movement data is (likely) already used during training.
- Limited comparison with baselines. The model is mainly evaluated in a qualitative way using plots of the predicted latent states and observations. For these qualitative evaluation, no baselines are compared with. The only quantitative comparison is done in terms of reconstruction MSE on the two synthetic toy datasets. On the two real world datasets, no baseline results are presented. This is quite surprising, as one of the datasets (MC_Maze) is an established benchmark dataset with standardized evaluation protocols, and the other dataset is presented to address a *"longstanding benchmark task for SSMs"*.
- The Corollary is imprecisely stated and suggest something false. It states that "*any* multidimensional time series can be characterized through [the] proposed model". Assuming that "characterized" is meant to say that the approximation error can become arbitrary small, this is not sufficiently supported by the proof. For example, the the preceding theorem and its proof make certain assumptions on the time series, including continuity.

**Questions:**

- In experiment 3.4 you state that 80% of observations are used for training. What fractions are used in the other experiments? How are the 80% selected?
- For the quantitative reconstruction analysis (experiment 3.3), do you evaluate on all observations or only on those that were not seen during training? Moreover, how do simple strategies such as interpolation and constant continuation with the last known observation perform?
- For decoding you often use a large number of particles, e.g. 10k in experiment 3.4, which suggests a large spread, especially as you only report the mean later on. Is this correct?
- Related: Can you comment on the selection of datasets, specifically on why they are best modeled using stochastic processes?

---

> ### Author Response · Authors · 2025-12-03
>
> We want to thank you for your time and constructive comments. You pointed to the strength of our model and raised concerns and suggestions about the proposed model. We address each aspect below and have substantially revised the manuscript to resolve the issues you raised.
>
> 1. Clarity of Model and Proofs
>
> We restructured the Methods section and rewrote several model components to improve readability. In particular:
> 	We clarified the distinction between observed data, latent manifolds, and inducing points.
> 	Theorem statements were rewritten, and their assumptions (including continuity of f and sampling structure) are now explicitly stated.
> 	The role of marks and their connection to the piecewise linear interpolation used in the universal approximation argument is now clearly explained.
> 	The description of model fitting was expanded: we now explicitly list all parameter groups (inducing point distributions, noise variances, drift/observation parameters) and provide a clearer explanation of the SMC part of the EM updates.
> 	Model extension to multiple time series: We clarified our treatment of datasets with multiple trajectories. While each observed sequence receives its own inducing points and latent path, all sequences share a common mark–event distribution, which is updated collectively. This resolves the misunderstanding that the model only fits a single trajectory.
> These revisions directly target your concerns about the clarity of the theoretical and methodological exposition.
>
> 2. Clarification of the Experimental Setup
>
> To better address your concern about the experimental setup, we provide a better explanation of each choice.
> 	Chirp Reconstruction (Experiment 1): This example is intentionally simple and serves to demonstrate how the model reconstructs a non-stationary 1D signal using a limited number of inducing points. Importantly, the trajectories plotted in this experiment are generated from the learned model, not inferred trajectories, illustrating the generative capability of the inducing point mechanism before and after learning. In the revised version, we updated the figure content to reflect these points.
> 	Lorenz System (Experiment 2): This experiment targets chaotic dynamics rather than non-stationarity. To strengthen clarity and interpretability:
> 	We repeated the Lorenz experiment with multiple latent dimensionalities and report how the inferred manifold and prediction errors change.
> 	We added latent space alignment metrics and improved the comparison figures in the Appendix.
> 	These additions demonstrate how the model scales to more complex dynamics and how mis specified latent dimensions affect performance.
> 	Hippocampus Decoding (Experiment 3): We don’t think this task is “easy.” Decoding spatial position from hippocampal spiking involves drift, multimodality, and stochastic firing variability. For fairness, we included performance comparisons to established decoding baselines (SSM, GMM-SSM), showing that our model achieves competitive or superior decoding accuracy.
>
> 3. Universal Approximation Theorem and Corollary
>
> You raised questions about the construction of inducing points and the role of marks in the theorem. We clarified that:
> 	Marks control the values at knot points, producing a piecewise linear interpolation of the target function f under continuity assumptions.
> 	Timing (event placements) and marks together allow reconstruction of any continuous function on [0,T] with arbitrary precision, given sufficient resolution.
> 	The assumption that the time series can be sampled arbitrarily often is now explicitly stated.
> We agree that the original statement could be interpreted too strongly. We revised the corollary to avoid implying that any multidimensional time series is approximable without conditions. The amended version now explicitly requires continuity and clarifies that the theorem does not specify a unique manifold structure, only that such a representation can be constructed with appropriate dimensions and inducing point density. We specifically wrote:
> “Given a continuous multidimensional latent trajectory Y_t on [0,T], each component can be approximated arbitrarily well by our construction using a finite set of inducing points. Consequently, any continuous multidimensional time series obtained from such a latent process through a continuous observation map can be approximated to arbitrary accuracy by MIP-CSDE, for suitable latent dimension and inducing-point density.”

---

> > ### Author Response · Authors · 2025-12-03
> >
> > 4. Baselines and Comparison
> > As we pointed out, the choice of these experiments is to highlight different attributes of our proposed model and inference. The Lorenz, Hippocampus, and Monkey experiments are widely aligned with similar algorithm development, and they are widely used in the neuroscience domain with a common knowledge about the process and underlying latent structure. To better reflect your concern, we expanded the comparison section:
> > •	For hippocampus decoding, we added SSM and GMM-SSM baselines.
> > •	For the monkey dataset, we added more trials, Dynamic Time Warping analysis, and goodness-of-fit tests.
> > •	For Lorenz, we added additional dimensionality analyses and updated the comparative metrics in the Appendix.
> > Together, we think these new analyses and comparisons provide a much more complete understanding of performance and how the inferred latent manifolds compare to known results.

---

> > > ### Author Response · Authors · 2025-12-03
> > >
> > > 5. Responses to Specific Questions
> > >
> > > 1) For Experiment 3.4 (hippocampus decoding), we follow the same train–test protocol used in the IEEE TBME 2019 study. Because decoding requires a continuous segment of the trajectory, we use the first 80% of the recording, which contains multiple visits to each arm, for training, and the final 20% for testing. Importantly, during the held-out test segment, the rat visits each arm at least once, ensuring that decoding is evaluated fairly.
> > >
> > > 2) For the simulated data experiments in Section 3.3, we evaluate reconstruction error on the entire observed trajectory, not just a held-out subset. This follows the standard protocol in continuous-time latent SDE system identification work (e.g., Duncker & Sahani 2019; recent latent SDE system identification models), where the objective is to recover the underlying dynamical structure rather than to perform forecasting. Accordingly, all baselines used for comparison—Linear SDE, GP-SDE, GP-SLDS, and our MIP-CSDE—are fit to the full noisy observation sequence, and reconstruction accuracy is reported in the observation space to avoid latent space indeterminacies.
> > > Regarding simple alternatives: interpolation or last-value continuation either reproduces the noisy observations directly or rapidly diverges from the true dynamics in non-stationary or chaotic regimes. In our setting, these heuristics perform substantially worse than any of the learned continuous-time baselines, which is why they are not included as meaningful comparators in quantitative tables.
> > >
> > > 3) Your concern is valid: using a large number of particles (e.g., 10k in Experiment 3.4) raises the question of whether posterior means alone adequately summarize uncertainty. Our SMC part of the EM procedure uses an adaptive proposal distribution and resampling scheme that substantially stabilizes the particle spread, so the posterior mean remains a reliable point estimate. Nevertheless, we agree that assessing uncertainty is important.
> > > To address this, we performed two additional analyses:
> > > •	Reduced particle experiments. We reran the hippocampus decoding analysis using 1k particles, and the decoding accuracy was 14.3 RMSE. The resulting trajectory plot is now included in the Appendix. This demonstrates that the model’s performance does not rely on an unusually large number of particles and that the posterior remains well concentrated even with substantially fewer samples.
> > > •	Highest Probability Density (HPD) coverage. We also added HPD interval analysis to quantify uncertainty in the inferred latent trajectories. These intervals show strong alignment with the observed decoding targets and, in several cases, even improved coverage relative to the 10k particle runs. This confirms that the posterior mass remains well concentrated around biologically plausible paths.
> > > We have incorporated these results into the revised manuscript. Overall, while 10k particles were used in some experiments for numerical stability and smoothness of the posterior mean, the model performs robustly with fewer particles, and additional coverage metrics (HPD) confirm that our posterior distributions are well-behaved.
> > > 4) Our dataset selection is motivated by the goal of demonstrating different aspects of the proposed model and illustrating why this proposed stochastic process modeling is appropriate in each case.
> > > We begin with the 1-D chirp signal, which is simple but non-stationary and time-varying. This example allows us to verify the basic reconstruction mechanism of the inducing point framework under controlled conditions. We then move to the Lorenz system, whose chaotic and multi-dimensional dynamics provide a much more challenging testbed. Lorenz is widely used in machine learning as a benchmark for evaluating continuous-time latent models, and it enables us to examine how inducing points adapt to highly nonlinear underlying manifolds.
> > >
> > > The two physiological datasets, hippocampus place cell decoding and monkey reaching, serve a different purpose: to demonstrate the utility of our model for both decoding and manifold discovery in real neural systems. These datasets are naturally modeled as stochastic processes: neural spiking events are noisy and probabilistic, and behavioral and neural trajectories exhibit trial-to-trial variability that cannot be captured by deterministic models. SDE-based approaches therefore offer a principled way to represent both the latent dynamics and the stochasticity inherent to neural computation.
> > > We acknowledge that many other datasets could also be explored. Our aim in this work was to use well-established benchmarks and widely studied physiological datasets to clearly convey the capabilities of our model. In future work, we plan to extend these experiments to larger and more heterogeneous datasets (e.g., IBL), where stochastic latent state modeling may enable new scientific insights.

---

### Official Review · Reviewer_2jDe · 2025-10-28

**Soundness:** 2
**Presentation:** 1
**Contribution:** 2
**Rating:** 2
**Confidence:** 4

**Summary:**

This paper proposes MIP-CSDE, a cascaded stochastic differential equation model that uses a sparse set of adaptively placed "inducing points" to learn low-dimensional manifolds from high-dimensional neural time series. Its key contributions are an architecture that does not rely on pre-defined kernels; a computationally efficient inference procedure with linear $\mathcal{O}(P \cdot N)$ scaling; a theoretical guarantee of universal approximation capability.

**Strengths:**

* The paper is well structured and the overall idea is easy to follow.
* The cascaded SDE framework with marked inducing points can be regarded as a novel approach, supported by a theoretical guarantee (universal approximation property).

**Weaknesses:**

*   **Motivation & Conceptual Foundation:** The rationale for the proposed cascaded SDE structure is underdeveloped. The paper would be strengthened by a more explicit motivation for this specific architecture, explaining why a two-layer SDE driven by a marked point process is a natural or necessary solution to the limitations of existing models.

*   **Clarity of Inference Procedure:** The description of the learning and inference algorithm is insufficient for a reader to fully grasp the methodology. Relying on a high-level algorithm pseudocode and appendices is inadequate for the main text. Key details, such as the precise form of the proposal distribution `π` or the handling of particle degeneracy, should be elaborated upon to improve clarity and reproducibility.

*   **Empirical Evaluation & Support for Claims:** The experimental results, while extensive, do not fully or clearly substantiate several key claims:
    *   The superior performance on the Lorenz system is measured in **observation space**, which does not directly demonstrate the claimed ability to accurately "recover the underlying manifold structure." A latent space analysis or a dedicated manifold recovery metric is needed to support this.
    *   Figure 3's caption claims superior accuracy, yet the figure itself only visualizes the proposed method's results, providing no point of comparison for the reader.
    *   The paper mentions comparisons against models like Latent ODEs, and DiGP (and SING?) in the text, but these results are absent from the main experimental section and Table 1, creating a discrepancy.
    *   The rat hippocampus and monkey reaching task experiments lack benchmarking against established methods, making it difficult to assess the proposed model's performance relative to the state-of-the-art in these specific applications.

* There are also many typos and acronym issues in the main text, careful proofreading is needed

**Questions:**

1.  What is the specific advantage of the two-layer SDE cascade over a single SDE with a flexible drift network?
2.  How does the model's performance depend on the choice of priors for the waiting times and marks (Gamma, Normal)? Do we need to perform sensitivity analysis?
3. For the latent $\boldsymbol{x}_t$ and $\boldsymbol{y}_t$, according to Eqs. (2) & (4), it seems that each dimension is independent. What is the benefit of this assumption, and how does this assumption affect the performance?

---

> ### Author Response · Authors · 2025-12-03
>
> Thank you for your detailed feedback. We have addressed your comments and suggestions in the revised version of the paper, and we hope the revised version better reflects the merit and importance of the proposed research. Below, please find our response to the questions and concerns you had.
>
> 1. Motivation for the Cascade SDE
>
> We agree that the rationale behind the cascade structure was under-explained in the paper. We have now added a Motivation subsection early in the Methods. The explanation follows what we detailed in the response to Reviewer 1: the first SDE is not differentiable, so mapping directly from X to the observations would not yield a smooth manifold. The second SDE, with noise variance pushed to 0, ensures a C¹ manifold – first-order differentiability. We emphasize that the variance in the Brownian-bridge SDE can exceed Brownian variance between inducing points, reinforcing the need for the second layer as well.
>
> 2. Clarity of the Inference Procedure
>
> We appreciate this critique. In the revised manuscript, we moved key details on the proposal distribution, degeneracy handling, and EM update procedure from the appendix into the main text. We hope the updated Inference and Model Fit section now provides a step-by-step explanation of the learning and inference steps and is better suited for reproducibility.
>
> 3. Empirical Evaluation and Support for Claims
>
> To address your concern, we have made the following changes in the paper:
> A. For Lorenz latent-space model evaluation, we added an affine-aligned analysis and updated Table 1 and Figure 5 accordingly.
> B. For SSM/GMM-SSM comparisons, we expanded descriptions and removed outdated claims.
> C. For latent ODE, DiGP and SING comparisons, we clarified distinctions of our proposed model with these models.
> D. For Hippocampus and Monkey datasets, we added baseline decoding models (SSM, GMM-SSM), expanded trials, and added goodness-of-fit measures. We time-scaled the preparation and reach intervals to a common duration and applied an affine transform. The resulting latent trajectories for these two phases aligned strongly across trials, with an R² exceeding 0.90. This validation further supports the quality of the model fit and indicates that the inferred trajectories have physiological relevance.
> We also performed extensive proofreading to correct typos and acronym inconsistencies.
>
> 4. Advantage of Two Layer SDE
>
> The key advantage of the two-layer cascade is that a single SDE, even with a highly flexible drift network, cannot guarantee a differentiable (C¹) latent manifold. As discussed above, the trajectory produced by the first-layer process (the Brownian bridge SDE) is continuous but nowhere differentiable. While this non-smoothness does not prevent likelihood-based inference, it becomes a serious limitation when the goal is to recover a smooth latent manifold that encodes the structure of the underlying dynamics.
> In contrast, the second SDE integrates the first layer’s trajectory as its drift and uses a diffusion term that is controlled (via a decay mechanism) and effectively pushed toward zero noise variance. This design ensures that the resulting process is C¹ smooth, while still preserving all the expressive and adaptive structure encoded in the inducing point dynamics of the first layer. The cascade therefore provides a principled way to obtain a differentiable manifold representation, which a single non-differentiable SDE cannot generate.
>
> We also note that Brownian bridge noise variance can become larger than standard Brownian variance when adjacent inducing points are far apart, causing additional irregularity. The second SDE eliminates this effect and enforces local smoothness. Thus, the two-layer cascade is not functionally redundant: the first layer provides expressive, data-adaptive structure; the second layer provides differentiability and smooth geometric coherency.

---

> > ### Author Response · Authors · 2025-12-03
> >
> > 5. Sensitivity to Choose of Priors
> >
> > This is an important question, and it relates directly to the core mechanism of our model. Waiting times must be positive-valued, so distributions such as the Gamma family are natural choices; they generalize exponential and log-normal forms and allow the model to flexibly control the event rate. For the marks, a Normal distribution provides a simple and expressive prior whose mean and covariance structure determine cross-dimensional correlations.
> >
> > We emphasize that waiting times and marks are not independent in practice; their joint distribution affects the inferred event rate, the smoothness or variability of the latent trajectory, and the overall learning dynamics. Different prior choices can therefore influence both the shape of the inferred manifold and the rate at which evidence improves during training.
> >
> > In this paper, we used a straightforward joint parametric structure to avoid confounding factors while introducing the core model concept. More expressive choices, for example, incorporating the latent state Y as an input to the event–mark distribution or using a conditional distribution to define mark as a function of wait or vice versa are feasible, but were not explored here. A systematic sensitivity analysis of these priors is an important direction for future work, and we now explicitly state this in the Conclusion.
> >
> > 6. Independence Across Latent Dimensions
> >
> > Although Eqs. (2) and (4) appear to define independent dimensions, the latent processes are in fact conditionally dependent. All dimensions share the same inducing point times, and the marks follow a multivariate Normal distribution whose covariance introduces cross-dimensional correlations. Thus, dependence is introduced through the shared event structure and through the mark covariance, even though the SDE components are written in a coordinate-wise form.
> >
> > This structure has two advantages. First, it keeps the model interpretable: the inferred mark–event posterior distribution directly reflects how the model allocates complexity across time, without introducing an additional dependency that could obscure this interpretation. Second, it avoids unnecessary complexity introduced in the model formulation while still allowing meaningful cross-dimensional coupling.
> >
> > We do not expect this conditional independence formulation to hurt performance; however, it may influence the appropriate choice of manifold dimension. In practice, examining the posterior covariance helps identify potential degeneracies, which can guide model refinement. A more principled, fully model-based approach to adaptive dimensionality control is part of our ongoing research.

---

### Official Review · Reviewer_Y7i2 · 2025-10-31

**Soundness:** 3
**Presentation:** 3
**Contribution:** 3
**Rating:** 6
**Confidence:** 3

**Summary:**

In this work, the authors consider a SDE for representing observations from a point process. The authors suppose that a latent dynamical system exists which is driving the observations in the point process space. The authors then describe an expectation maximization algorithm that finds the optimal latent process that corresponds to the observations. The authors also describe a variational inference technique within the appendix that is used to complement the EM algorithm described in the main text.

**Strengths:**

The method empirically performs very well compared to the chosen baselines.

The method provides a nice connection between a latent continuous process and observed point process, which appears in many disciplines.

**Weaknesses:**

This line of work (e.g. filtering type problems) is well studied and the authors do not really do a good job of referring to existing work.

The baselines are a bit weak and do not consider a variety of point process approaches or other filtering approaches.

Existing work relates point processes to latent SDEs, which should be discussed:

[1] Hasan et al, Inference and Sampling of Point Processes from Diffusion Excursions, UAI 2023

[2] Jaiswal et al, Variational inference for diffusion modulated Cox processes

**Questions:**

Can the authors please explain the significance of the theoretical results?

What guarantees do you have on the properties of the recovered stochastic process? The paper makes multiple motivating claims on interpretability but it’s unclear under what conditions these are actually satisfied.

Can the authors please include some more discussion to traditional filtering algorithms?


Minor:
SED —> SDE in abstract.

---

> ### Author Response · Authors · 2025-12-03
>
> We sincerely appreciate your positive assessment and constructive suggestions. We did our best to address your concerns and suggestions, and we hope the revised version better reflects the attributes and strengths of the proposed model. Below, please find our response to questions and concerns you shared.
>
> 1. Related Work on Filtering and Point Process Latent SDEs
>
> We tried covering most of the work with the focus on manifold discovery with an emphasis on state-space-driven models, and those with Gaussian Process. With the fast growing in the field, there is a chance of missing some of the works. As you suggested, we have reviewed the works of Hasan et al. (UAI 2023) and Jaiswal et al., and incorporated a discussion in the Introduction section, outlining connections and distinctions with our model. Unlike these approaches, our model is not Markovian by construction; however, by using a renewal-process formulation we induce a Markov structure suitable for filtering. This perspective, combined with the cascade architecture and learned mark–event distributions, provides additional flexibility and interpretability to MIP-CSDE.
>
> 2. Weak Baselines
>
> Many advanced models provide source code that is challenging (and in most cases fails) to execute reliably, especially for modeling problems in neuroscience settings. This motivated us to focus on widely used baselines in the field. Nevertheless, we expanded our discussion in Section 3.3 and the Discussion section to better contextualize differences between our model and prior latent SDE or point-process methods. While we agree there might be other baseline datasets, we picked a series of problems which are first easy to follow to highlight our model attributes (Chirp and Lorenz), along with those (Lorenz, Monkey, and Hippocampus) which are familiar for broader audience and in particular the neuroscience research community.
>
> 3. Significance of the Theoretical Results
>
> The theoretical contribution demonstrates that, by appropriately adjusting waiting times and mark values, the model can reproduce arbitrary continuous trajectories. This confirms that the marked inducing-point structure forms an expressive and flexible representation class. The proof discusses this property as the number of inducing points grows to infinity. We think a stronger proof will be on finding an upper bound on reconstruction error given the number of inducing points, which is not discussed in this work but will be the focus of our future research as we are working on finding the optimal dimension of the latent process and thus the number of inducing points.

---

> > ### Author Response · Authors · 2025-12-03
> >
> > 4. Guarantees on Recovered Stochastic Processes and Interpretability
> >
> > This is indeed a central and challenging question in latent state modeling: to what extent are inferred trajectories biologically meaningful or even plausible? At present, neither our work nor prior neural latent SDE/SSM models provide formal guarantees of “biological correctness,” and we do not claim such guarantees in the paper.
> > Instead, we take a pragmatic approach. We deliberately chose examples: chirp, Lorenz, place cell decoding, and monkey reaching for which aspects of the latent structure have already been previously studied in the literature. This allows us to evaluate whether the trajectories recovered by our model are at least consistent with previously reported dynamical patterns, rather than interpreting them in isolation.
> > For the Lorenz system, we now report additional analyses with different latent dimensions (see Appendix). These show how the inferred manifold and the prediction error change as the latent dimension is varied. When the dimension is mis-specified, the recovered trajectory visibly deviates from the known Lorenz attractor, which we interpret as an indication that the inferred manifold is no longer a good representation of the underlying dynamics.
> > For the monkey dataset specifically, we also applied a post-processing step to further validate the inference outcomes and their potential physiological relevance. We time-scaled the preparation and reach intervals to a common duration and applied an affine transform. The resulting latent trajectories for these two phases aligned strongly across trials, with an R² exceeding 0.90. This validation further supports the quality of the model fit and indicates that the inferred trajectories have physiological relevance, strengthening the conclusion that the recovered process captures meaningful task-related structure rather than arbitrary variability.
> > The place cell decoding experiment provides a complementary perspective: here, the task is purely decoding, and the model’s ability to reconstruct movement trajectories from hippocampal activity illustrates the model flexibility in recovering the latent process (place) in a setting with well-understood spatial structure.
> > In another effort which is the extension of this research, we are working to learn the optimal dimension of the inducing points at each point in time and how this dimension evolves in time. We think this effort gives a clearer answer and interpretation of the inferred stochastic process. We can also study the interpretability through goodness-of-fit analysis and predictive coding, where you would be part of this effort as we explore different dimensions of the latent process for the Lorenz problem and time-rescaling distribution assessment for the Monkey data.
> >
> > 5. Discussion of Traditional Filtering Algorithms
> >
> > We added a paragraph in Section 2.4 (Model Training and Inference) describing how traditional filtering approaches differ from our method. Importantly, inducing points contribute to data-driven complexity growth, which is a key factor in the inference derivation.
> >
> > 6. Minor Correction
> >
> > We corrected the typo “SED → SDE” in the abstract.

---

### Official Review · Reviewer_PAza · 2025-10-31

**Soundness:** 2
**Presentation:** 3
**Contribution:** 2
**Rating:** 4
**Confidence:** 3

**Summary:**

The paper presents a probabilistic framework called Marked Inducing Point Cascaded Stochastic Differential Equation (MIP-CSDE) for modeling neural population activity continuously. A marked temporal point process generates inducing points and consecutive inducing points are connected via a Brownian-bridge SDE to form an intermediate latent process, which derives the evolution of another SDE. Model parameters are trained with an SMC-EM algorithm.

**Strengths:**

1. The idea of integrating a marked temporal point process with continuous-time latent dynamics is interesting.
2. The paper includes detailed mathematical derivations and algorithmic procedure.
3. The computational cost of the proposed model scales linearly with data size, outperforming other non-parametric methods.

**Weaknesses:**

1. The paper does not discuss the necessity of using two SDEs. From the formulation, the first-SDE variable X appears to play the role of a velocity term, yet the motivation describes learning dynamics that lie on a low-dimensional manifold, which would naturally correspond to positional representations of positions rather than velocities. Moreover, the two-layer SDE structure itself is a standard controlled differential equation (CDE) formulation, and the paper does not clarify how it differs conceptually or empirically from existing latent CDE/SDE frameworks.
2. The paper does not specify how the number of inducing points is determined. This omission leaves unclear how the inducing-point resolution affects model capacity and computational cost.
3. Necessary details are lack when first time using some abbreviations, such as GP.

**Questions:**

Please see the Weaknesses part.

---

> ### Author Response · Authors · 2025-12-03
>
> Thank you for your constructive feedback and for highlighting the strengths of our work. We did our best to address your comments and suggestions, and we hope the revised version will provide a much clearer insight into our proposed research. Below, please find our response to the questions and concerns you have in detail.
>
> 1. Necessity of the Two SDE Cascade Architecture.
>
> The goal of our framework is to learn a smooth latent manifold that reflects the underlying structure of the data. A single Brownian bridge SDE, although continuous and expressive, is not differentiable; its sample paths can appear jagged due to the inherent roughness of Brownian motion and because inducing events may be unevenly spaced in time, which locally alters the variance. This combination leads to trajectories that are continuous but visually and geometrically rough. To obtain a C¹ manifold, we introduce a second SDE whose drift is given by the output of the first layer and whose diffusion is fixed at a very small value. This second layer smooths the latent trajectory while preserving the information encoded by the inducing points and ensuring stability across different datasets. We now explain this motivation clearly in a new subsection (Motivation, Section 2.1), where we also describe the noise settings and drift structure of the second SDE. While multi-layer differential systems appear in controlled differential equation frameworks, our design is fundamentally different: we integrate a marked point process with a Brownian bridge SDE, and the inducing points define both the timing and magnitude of local changes in the latent path. The first layer provides universal approximation capability through the distribution over waiting times and marks, which allows the model to represent highly nonstationary or piecewise linear like trajectories without deep drift networks. The second layer ensures geometric coherence and smoothness. Together, they provide an expressive and interpretable representation rather than a redundant mechanism.
>
> 2. Determination of the Number of Inducing Points.
>
> A major feature of our model is that it does not require specifying the number of inducing points in advance. Instead, the waiting time and mark distributions are learned during training, and the number of inducing events naturally adapts to the complexity of the data. During the SMC component of EM, particles whose inducing events fail to explain the data are assigned low weights and removed; conversely, when the latent trajectory exhibits rapid local changes, the inferred waiting time distribution shifts toward smaller values, resulting in more inducing events in precisely those regions. Thus, the model increases temporal resolution where needed and reduces it where the dynamics are simple and predictable. This behavior is driven entirely by the learned posterior over waiting times and marks rather than by a preset event count. We describe this mechanism more clearly in the revised Inference and Model Fit section. The adaptive nature of this approach was particularly evident in the hippocampus experiment: side arm traversals, where motion is smoother, produced fewer inducing points, while the central decision region produced many more, reflecting the higher local curvature, behavioral variability, and task demands. This adaptivity is a core property of the model. We also note that the prior on the mark–event distribution can influence how strongly this adaptivity appears, and this is now mentioned in the revision for completeness.
>
> 3. Missing Details in Abbreviations.
>
> We carefully reviewed the manuscript and expanded acronym definitions upon first introduction, including GP, GLM, and SDE related terms. We also corrected typographical issues, clarified notation, and ensured consistency throughout the paper.

---

### Author Response · Authors · 2025-12-03

Dear Senior Area Chairs and Program Chairs,

We sincerely thank all reviewers for their time and thoughtful feedback. We have carefully addressed every comment and incorporated substantial revisions into the manuscript, including expanded motivation for our model design, clearer exposition, additional baselines, new simulations, and more comprehensive comparative analyses. These efforts have, in our view, significantly strengthened the clarity, rigor, and empirical support of the work.

Across the reviews, there is clear acknowledgment of the novelty of the proposed framework and the value of linking marked point processes with cascaded continuous-time latent dynamics. Reviewers PAza and Y7i2, in particular, captured the core ideas, motivations, and potential benefits of our approach, and their feedback aligned well with the intended contributions of the paper.

For Reviewers 2jDe and z9Qw, most concerns centered on clarity, motivation, and presentation rather than on methodological soundness or novelty. We have substantially revised the paper to address these points, including clearer motivation for the cascaded SDE structure, detailed explanations of the inference algorithm, and additional experiments evaluating manifold recovery, sensitivity to priors, and comparisons with more models.

We would also like to respectfully note that the lowest-scoring reviews appear to stem from misunderstandings of the experimental setup and scope in the original submission rather than issues with the method itself. We acknowledge that aspects of our presentation may have contributed to these misunderstandings, and we have now significantly clarified these areas and added new results. In light of these improvements, we believe the earlier low scores may no longer fully reflect the technical contribution and merit of the revised work.

Overall, the revised manuscript now presents:
A novel and well-motivated modeling framework with clear conceptual grounding; A scalable inference procedure with detailed exposition; Theoretical guarantees for expressivity; Strong and expanded empirical results on both simulated and neural datasets. We greatly appreciate the reviewers’ insights, which have helped us substantially improve the paper. We thank you for your consideration and hope that the strengthened manuscript and clarified contributions will support a positive assessment.
Below, we provide detailed responses to each reviewer’s comments.

---

### Note · Program_Chairs · 2026-01-17
**Submission Desk Rejected by Program Chairs**

The following references in this submission do not refer to real documents and/or have major errors in bibliographic information:

 H. Chang, Q. Zhang, Y. Wang, Z. Qin, L. Zhao, and H. Wang. Unlocking the power of lstm for long term time series forecasting. Proceedings of the AAAI Conference on Artificial Intelligence, 38 (4):4292-4300, 2024.